# AH-UGC: Adaptive and Heterogeneous-Universal Graph Coarsening

## Abstract

**Graph Coarsening (GC)** is a prominent graph reduction technique that compresses large graphs to enable efficient learning and inference. However, existing GC methods generate only one coarsened graph per run and must recompute from scratch for each new coarsening ratio, resulting in unnecessary overhead. Moreover, most prior approaches are tailored to *homogeneous* graphs and fail to accommodate the semantic constraints of *heterogeneous* graphs, which comprise multiple node and edge types. To overcome these limitations, we introduce a novel framework that combines Locality-Sensitive Hashing (LSH) with Consistent Hashing to enable *adaptive graph coarsening*. Leveraging hashing techniques, our method is inherently fast and scalable. For heterogeneous graphs, we propose a *type-isolated coarsening* strategy that ensures semantic consistency by restricting merges to nodes of the same type. Our approach is the first unified framework to support both adaptive and heterogeneous coarsening. Extensive evaluations on 23 real-world datasets—including homophilic, heterophilic, homogeneous, and heterogeneous graphs demonstrate that our method achieves superior scalability while preserving the structural and semantic integrity of the original graph. Our code is available here.

## 1 Introduction

Graphs are ubiquitous and have emerged as a fundamental data structure in numerous real-world applications [1–3]. Broadly, graphs can be categorized into two types: (a) *Homogeneous graphs* [4–6], which consist of a single type of nodes and edges. For instance, in a homogeneous citation graph, all nodes represent papers, and all edges represent the "cite" relation between them; (b) *Heterogeneous graphs* [7–9], which involve multiple types of nodes and/or edges, enabling the modeling of richer and more realistic interactions. For example, in a recommendation system, a heterogeneous graph may contain nodes of different types, such as users, items, and categories, and edge types such as "(user, buys, item)", "(user, views, item)", and "(item, belongs-to, category)". Although many real-world datasets are inherently heterogeneous, early research in graph machine learning predominantly focused on homogeneous graphs due to their modeling simplicity, availability of standardized benchmarks, and theoretical tractability [10, 11]. However, the limitations of homogeneous representations in capturing rich semantic information have shifted attention toward heterogeneous graph modeling [8, 12].

As real-world networks continue to grow rapidly in size and complexity, large-scale graphs have become increasingly common across various domains [1, 13–15]. This surge in scale poses significant computational and memory challenges for learning and inference tasks on such graphs. This underscores the growing importance of developing efficient and effective methodologies for processing large-scale graph data. To address the issue, an expanding line of research investigates graph reduction methods that compress structures without compromising essential properties. Most existing graph reduction techniques, including pooling [16], sampling-based [17], condensation [18], and

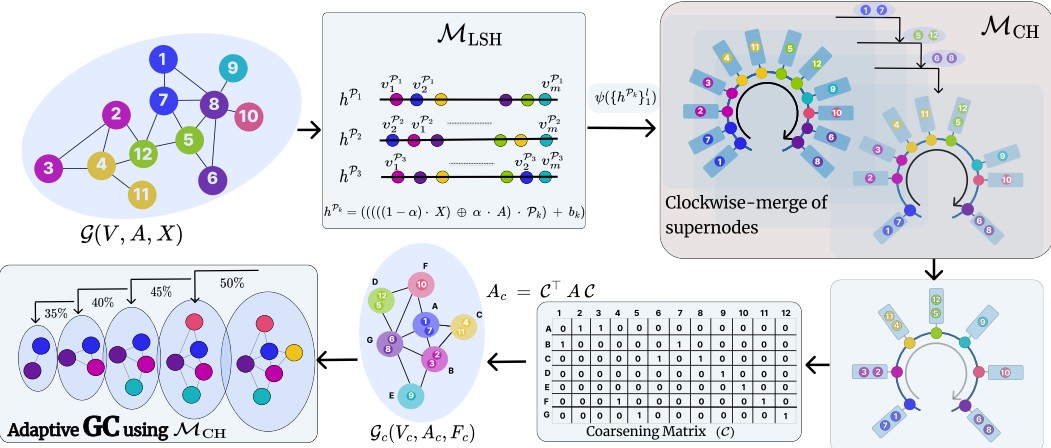

Figure 1: AH-UGC consists of three modules: (a) $\mathcal{M}_{\text{LSH}}$ constructs an augmented feature matrix by combining node features and structural context using a heterophily-aware factor $\alpha$, enabling support for both homophilic and heterophilic graphs. Inspired by UGC [4], we use LSH projections to compute node hash indices via $\psi(h^{\mathcal{P}k^l}_1)$ (see Section 3); (b) $\mathcal{M}_{\text{CH}}$ applies consistent hashing to merge nodes clockwise based on a target coarsening ratio $r$, yielding the coarsening matrix $\mathcal{C}$; (c) the coarsened graph $\mathcal{G}_c$ is obtained via $A_c = \mathcal{C}^\top A \mathcal{C}$. The framework is inherently adaptive— i.e., once an intermediate coarsening is obtained, further reduction can be applied incrementally using $\mathcal{M}_{\text{CH}}$ and already calculated coarsening matrix $\mathcal{C}$, enabling efficient multi-resolution processing.

coarsening-based methods [4, 19, 20]. Coarsening methods have demonstrated effectiveness in preserving structural and semantic information [4, 19, 20], this study focuses on graph coarsening (GC) as the primary reduction strategy. Despite advancements in existing GC frameworks, two key challenges remain:

- **Lack of "Adaptive Reduction".** Many applications, such as interactive visualization and real-time recommendations, benefit from multi-resolution graph representations. These scenarios often require dynamically adjusting the coarsening ratio based on user interaction or task demands. However, most existing methods generate a single fixed-size coarsened graph and must recompute from scratch for each new ratio, incurring high overhead. This highlights the need for adaptive coarsening frameworks that enable efficient, progressive refinement without redundant computation.
- **Lack of "Heterogeneous Graph Coarsening" Framework.** Existing methods typically assume homogeneous node types, making them unsuitable for heterogeneous graphs with semantically distinct nodes. This can result in invalid supernodes for example, merging an *author* with a *paper* node in a citation graph thus violating type semantics. Moreover, node types often have different feature dimensions, which standard coarsening techniques are not designed to handle.

**Key Contribution.** To address the dual challenges of adaptive reduction and heterogeneous GC, we propose **AH-UGC**, a unified framework for Adaptive and Heterogeneous Universal Graph Coarsening. We integrate locality-sensitive hashing (LSH) [4, 21, 22] with consistent hashing (CH) [23, 24]. While LSH ensures that similar nodes are coarsened together based on their features and connectivity, CH—a technique originally developed for load balancing—enables us to design a coarsening process that supports multi-level adaptive coarsening without reprocessing the full graph. To handle heterogeneous graphs, AH-UGC enforces *type-isolated coarsening*, wherein nodes are first grouped by their types, and coarsening is applied independently within each type group. This ensures that nodes and edges of incompatible types are never merged, preserving the semantic structure of the original heterogeneous graph. Additionally, AH-UGC is naturally suited for streaming or evolving graph settings, where new nodes and edges arrive over time. Our LSH- and CH-based method allows new nodes to be integrated into the existing coarsened structure with minimal recomputation. To summarize, **AH-UGC** is a general-purpose graph coarsening framework that supports *adaptive, streaming, expanding, heterophilic, and heterogeneous graphs*.

## 2 Background

**Definition 2.1 (Graph)** *A graph is represented as $\mathcal{G}(V, A, X)$, where $V = \{v_1, \ldots, v_N\}$ is the set of $N$ nodes, $A \in \mathbb{R}^{N \times N}$ is the adjacency matrix, and $X \in \mathbb{R}^{N \times \widetilde{d}}$ is the node fea-*

ture matrix with each row $X_i \in \mathbb{R}^{\widetilde{d}}$ denoting the feature vector of node $v_i$. An edge between nodes $v_i$ and $v_j$ is indicated by $A_{ij} > 0$. Let $D \in \mathbb{R}^{N \times N}$ be the degree matrix with $D_{ii} = \sum_j A_{ij}$, and let $L = D - A$ denote the unnormalized Laplacian matrix. $L \in S_L$, where $S_L = \left\{ L \in \mathbb{R}^{N \times N} \,\middle|\, L_{ij} = L_{ji} \leq 0 \, for \, i \neq j; \; L_{ii} = -\sum_{j \neq i} L_{ij} \right\}$. For $i \neq j$, the matrices are related by $A_{ij} = -L_{ij}$, and $A_{ii} = 0$. Hence, the graph $\mathcal{G}(V, A, X)$ may equivalently be denoted $\mathcal{G}(L, X)$, and we use either form as contextually appropriate.

**Definition 2.2** *A heterogeneous graph can be represented in two equivalent forms, with either representation utilized as required within the paper.*

- ***Entity-based:*** *A heterogeneous graph extends the standard graph structure by incorporating multiple types of nodes and/or edges. Formally, a heterogeneous graph is defined as $\mathcal{G}(V, E, \Phi, \Psi)$, where $\Phi : V \to \mathcal{T}_V$ and $\Psi : E \to \mathcal{T}_E$ are node-type and edge-type mapping functions, respectively [9]. Here, $\mathcal{T}_V$ and $\mathcal{T}_E$ denote the sets of possible node types and edge types. When the total number of node types $|\mathcal{T}_V|$ and edge types $|\mathcal{T}_E|$ is equal to 1, the graph degenerates into a standard homogeneous graph (Definition 2.1).*
- ***Type-based:*** *Alternatively, a heterogeneous graph can be described as $\mathcal{G}\left(\{X_{(node\_type)}\}, \{A_{(edge\_type)}\}, \{y_{(target\_type)}\}\right)$, where feature matrices $X$, adjacency matrices $A$, and target labels $y$ are grouped and indexed by their corresponding node, edge, and target types [25].*

**Definition 2.3** *Following [4, 19, 20], The **G**raph **C**oarsening (GC) problem involves learning a coarsening matrix $\mathcal{C} \in \mathbb{R}^{N \times n}$, which linearly maps nodes from the original graph $\mathcal{G}$ to a reduced graph $\mathcal{G}_c$, i.e., $V \to \widetilde{V}$. This linear mapping should ensure that similar nodes in $\mathcal{G}$ are grouped into the same super-node in $\mathcal{G}_c$, such that the coarsened feature matrix is given by $\widetilde{X} = \mathcal{C}^T X$. Each non-zero entry $\mathcal{C}_{ij}$ denotes the assignment of node $v_i$ to super-node $\widetilde{v}_j$. The matrix $\mathcal{C}$ must satisfy the following structural constraints:*

$$S = \{\mathcal{C} \in \mathbb{R}^{N \times n}, \; \mathcal{C}_{ij} \in \{0, 1\}, \; \|\mathcal{C}_i\| = 1, \; \langle \mathcal{C}_i^T, \mathcal{C}_j^T \rangle = 0 \; \forall i \neq j, \; \langle \mathcal{C}_l, \mathcal{C}_l \rangle = d_{\widetilde{V}_l}, \; \|\mathcal{C}_i^T\|_0 \geq 1\}$$

*where $d_{\widetilde{V}_l}$ means the number of nodes in the $l^{th}$-supernode. The condition $\langle \mathcal{C}_i^T, \mathcal{C}_j^T \rangle = 0$ ensures that each node of $\mathcal{G}$ is mapped to a unique super-node. The constraint $\|\mathcal{C}_i^T\|_0 \geq 1$ requires that each super-node contains at least one node.*

## 2.1 Problem formulation and Related Work

We formalize the problem through two key objectives: Goal 1. Adaptive Coarsening and Goal 2. Graph Coarsening for Heterogeneous Graphs.

**Goal 1.** The objective is to compute multiple coarsened graphs $\{\mathcal{G}_c^{(r)}\}_{r=1}^R$ from input graph $\mathcal{G}(V, A, X)$, where each $\mathcal{G}_c^{(r)}$ corresponds to a target coarsening ratio $r \in (0, 1]$, without recomputing from scratch for each resolution. Formally, the goal is to construct a family of coarsening matrices $\{\mathcal{C}^{(r)} \in \mathbb{R}^{N \times n^{(r)}}\}$ such that

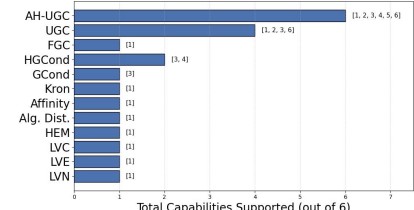

$$\widetilde{X}^{(r)} = (\mathcal{C}^{(r)})^\top X, \quad \widetilde{A}^{(r)} = (\mathcal{C}^{(r)})^\top A \mathcal{C}^{(r)},$$

Figure 2: Comparison of capability support across existing GC methods.

with the constraint that all $\mathcal{C}^{(r)}$ are derived from a single, shared projection $s = \text{HASH}(X)$, thereby ensuring consistency across coarsening levels and enabling adaptive GC.

**Goal 2.** The objective is to learn a coarsening matrix $\mathcal{C} \in \mathbb{R}^{N \times n}$, such that the resulting coarsened graph $\mathcal{G}_c(\widetilde{V}, \widetilde{E}, \widetilde{\Phi}, \widetilde{\Psi})$ satisfies the following constraints:

$$\widetilde{\Phi}(\widetilde{v}_j) = \Phi(v_i), \quad \forall \widetilde{v}_j \in \widetilde{V}, \forall v_i \in \pi^{-1}(\widetilde{v}_j),$$
$$\widetilde{\Psi}(\widetilde{v}_j, \widetilde{v}_k) \in \mathcal{T}_E \quad \text{only if} \quad \exists (v_i, v_l) \in E \text{ s.t. } \pi(v_i) = \widetilde{v}_j, \; \pi(v_l) = \widetilde{v}_k,$$

where $\pi : V \to \widetilde{V}$ is the node-to-supernode mapping induced by $\mathcal{C}$. These constraints guarantee that: a) nodes of different types are not merged into the same supernode, and b) edge types between supernodes are consistent with the original heterogeneous schema.

**Related Work.** Graph reduction methods have been extensively studied and can be broadly categorized into optimization-based and GNN-based approaches. Among optimization-driven heuristics, Loukas's spectral coarsening methods [20] including edge-based (LVE) and neighborhood-based (LVN) variants aim to preserve the spectral properties of the original graph. Other techniques, such as Heavy Edge Matching (HEM)[17, 26], Algebraic Distance[27], Affinity [28], and Kron reduction [29], rely on topological heuristics or structural similarity principles. FGC [19] incorporates node features to learn a feature-aware reduction matrix. Despite their diverse designs, a common drawback of these methods is that they are computationally demanding, often with time complexities ranging from $\mathcal{O}(n^2)$ to $\mathcal{O}(n^3)$, and are not well suited for large-scale or adaptive graph reduction settings. UGC [4], a recent LSH-based framework, addresses these challenges by operating in linear time and supporting heterophilic graphs. However, it produces only a single coarsened graph and must recompute reductions for different coarsening levels, limiting its adaptability. GNN-based condensation methods like GCond [30] and SFGC [31] learn synthetic graphs through gradient matching but require full supervision, are model-specific, and lack scalability. HGCond [25] is the only approach designed for heterogeneous graphs, yet it inherits the inefficiencies of condensation-based techniques.

While some methods are model-agnostic, others offer partial support for heterophilic or streaming graphs. Yet, no existing approach simultaneously addresses all these challenges—model-agnosticism, adaptability, and support for heterophilic, heterogeneous, and streaming graphs. As illustrated in Figure 2, HA-UGC is the first framework to meet all six criteria comprehensively. For details on LSH and consistent hashing, see Appendix B.

## 3 The Proposed Framework: Adaptive and Heterogeneous Universal Graph Coarsening

In this section we propose our framework AH-UGC to address the issues of adaptive and heterogeneous graph coarsening. Figure 1 shows the outline of AH-UGC.

### 3.1 Adaptive Graph Coarsening(Goal 1)

The AH-UGC pipeline closely follows the recently proposed structure of UGC but incorporates consistent hashing principles to enable adaptive i.e., multi-level coarsening. Our framework introduces an innovative and flexible approach to graph coarsening that removes the UGC's dependency on fixed bin widths and enables the generation of multiple coarsened graphs. Similar to UGC [4], AH-UGC employs an augmented representation to jointly encode both node attributes and graph topology. For a given graph $\mathcal{G}(V, A, X)$, we compute a heterophily factor $\alpha \in [0, 1]$, which quantifies the relative emphasis on structural information based on label agreement between connected nodes i.e., $\alpha = \frac{|\{(v,u) \in E : y_v = y_u\}|}{|E|}$. This factor is then used to blend node features $X_i$ and adjacency vectors $A_i$. For each node $v_i$ we calculate $F_i = (1 - \alpha) \cdot X_i \oplus \alpha \cdot A_i$ where $\oplus$ denotes concatenation. This hybrid representation ensures that both local attribute similarity and topological proximity are captured before the coarsening process. Importantly, this design enables our framework to handle heterophilic graphs robustly by incorporating structural properties beyond mere feature similarity.

**Adaptive Coarsening via Consistent and LSH Hashing.** Let $F_i \in \mathbb{R}^d$ denote the augmented feature vector for node $v_i$. AH-UGC applies $l$ random projection functions using a projection matrix $\mathcal{W} \in \mathbb{R}^{d \times l}$ and bias vector $b \in \mathbb{R}^l$, both sampled from a $p$-stable distribution [32]. The scalar hash score for each projection for $i^{th}$ node is given by:

$$s_i^k = \mathcal{W}_k \cdot F_i + b_k, \quad \forall k \in \{1, \ldots, l\}$$

UGC relies on a bin-width parameter $(r)$ to control the coarsening ratio $(R)$, but determining appropriate bin-widths for different target ratios can be computationally expensive. In contrast, AH-UGC eliminates the need for bin width by leveraging consistent hashing. Once the hash scores $(s_i)$ across projections are computed, AH-UGC enables efficient construction of coarsened graphs at multiple coarsening ratios without requiring reprocessing, making it well-suited for adaptive settings. We define an AGGREGATE function to combine projection scores across multiple random projectors. For each node $i$, the final score $s_i$ is computed as:

$$s_i = \text{AGGREGATE}\left(\{s_i^k\}_{k=1}^l\right) = \frac{1}{l} \sum_{k=1}^l s_i^k$$

Alternative aggregation functions such as max, median, or weighted averaging can also be used, depending on the design objectives. After computing the scalar hash scores $\{s_i\}$ for all nodes $v_i \in V$,

we sort the nodes in increasing order of $s_i$ to form an ordered list $\mathcal{L}$, represented as a list of super-node and mapped nodes: $\mathcal{L} = [\{u_1 : \{v_1\}\}, \{u_2 : \{v_2\}\}, \ldots, \{u_n : \{v_n\}\}]$, where each key $u_j$ denotes a super-node index, and the associated value is the set of nodes currently assigned to that super-node. Initially, each node is its own super-node, and the number of super-nodes is $|V_c^{(0)}| = |V|$. At each iteration $t$, a super-node $u_j$ is randomly selected from the current list $\mathcal{L}^{(t)}$ and merged with its immediate clockwise neighbor $u_{j+1}$. The updated super-node entry is given by:

$$\mathcal{L}^{(t+1)}[j] = \{u_j : \mathcal{L}^{(t)}[u_j] \cup \mathcal{L}^{(t)}[u_{j+1}]\},$$

followed by the removal of $u_{j+1}$ from the list. This reduces the number of super-nodes by one: $|V_c^{(t+1)}| = |V_c^{(t)}| - 1$. The process is repeated until the desired coarsening ratio is reached: $R = \frac{|V_c|}{|V|}$. Furthermore, this coarsening strategy is inherently adaptive, enabling transitions between any two coarsening ratios $R \to T$ directly from the sorted list without reprocessing.

Since the list $\mathcal{L}$ is constructed using locality-sensitive hashing (LSH) principles [32], similar nodes are positioned adjacently. Through Theorem 3.1 and Lemma 1, we show that the clockwise merging operations in Consistent Hashing (CH) are locality-aware and effectively preserve feature similarity.

**Theorem 3.1** *Let $x, y \in \mathbb{R}^d$, and let the projection function be defined as: $h(x) = \sum_{j=1}^{\ell} r_j^\top x$, $r_j \sim \mathcal{N}(0, I_d)$ i.i.d. Then the difference $h(x) - h(y) \sim \mathcal{N}(0, \ell \|x - y\|^2)$, and for any $\varepsilon > 0$:*

$$\Pr\left[|h(x) - h(y)| \le \varepsilon\right] = \mathrm{erf}\left(\frac{\varepsilon}{\sqrt{2\ell}\|x - y\|}\right)$$

**Proof:** The proof is deferred in Appendix D.
This gives the probability that two nodes, initially close in the feature space, are projected within an $\epsilon$-range in the projection space.

**Lemma 1** *Let $x, y, z \in \mathbb{R}^d$, with $\|x - y\| \ll \|x - z\|$. Then the probability that a distant point $z$ lies between $x$ and $y$ after projection is:*

$$\Pr[h(x) < h(z) < h(y)] \le \Phi\left(\frac{\|x - y\|}{\sqrt{\ell}\|x - z\|}\right)$$

*where $\Phi$ is the cumulative distribution function (CDF) of the standard normal distribution. This result ensures that distant nodes rarely interrupt merge candidates that are close in feature space, preserving the structural consistency of coarsened regions.*

**Remark 1** *Our framework also supports de-coarsening i.e., given the final sorted list and merge history, the graph can be reconstructed to finer resolutions by reversing the merging process. However, in this work, we restrict our focus to the coarsening direction only.*

**Construction of Coarsening Matrix $\mathcal{C}$.** Given the score-based node assignments $\pi : V \to \widetilde{V}$, where $\pi[v_i]$ is the super-node index of $v_i$, the binary coarsening matrix $\mathcal{C} \in \{0, 1\}^{N \times n}$ is defined such that $\mathcal{C}_{ij} = 1$ if $\pi[v_i] = \widetilde{v}_j$, and $\mathcal{C}_{ij} = 0$ otherwise. Each entry $\mathcal{C}_{ij}$ of the coarsening matrix is set to 1 if node $v_i$ is assigned to super-node $\widetilde{v}_j$. Since each node receives a unique hash value $h_i$, it is exclusively mapped to a single super-node. This one-to-one assignment guarantees that every super-node has at least one associated node. As a result, each row of $\mathcal{C}$ contains exactly one non-zero entry, ensuring that its columns are mutually orthogonal. The matrix $\mathcal{C}$ therefore adheres to the structural properties defined in Equation 2.3. The adaptiveness of $\mathcal{C}$ stems from its sensitivity to local projection scores rather than fixed bin constraints.

**Construction of the Coarsened Graph $\mathcal{G}_c$.** The final coarsened graph $\mathcal{G}_c = (\widetilde{V}, \widetilde{A}, \widetilde{F})$ is constructed from the coarsening matrix $\mathcal{C}$. Two super-nodes $\widetilde{v}_i$ and $\widetilde{v}_j$ are connected if there exists at least one edge $(u, v) \in E$ with $u \in \pi^{-1}(\widetilde{v}_i)$ and $v \in \pi^{-1}(\widetilde{v}_j)$. The weighted adjacency matrix is obtained via matrix multiplication: $\widetilde{A} = \mathcal{C}^T A \mathcal{C}$. The super-node features are computed as the average of the features of the original nodes merged into the super-node: $\widetilde{F}_i = \frac{1}{|\pi^{-1}(\widetilde{v}_i)|} \sum_{u \in \pi^{-1}(\widetilde{v}_i)} F_u$. This ensures that the coarsened representation preserves the aggregate semantic and structural content of its constituent nodes. Since each super-edge aggregates multiple edges from the original graph, $\widetilde{A}$ is significantly sparser than $A$, leading to lower memory and computation requirements downstream. Algorithm 1 in Appendix G outlines the sequence of steps in our AH-UGC framework.

## 3.2 Heterogeneous Graph Coarsening

In this section, we present AH-UGC's capability to handle heterogeneous graphs. Given a heterogeneous graph,

$$\mathcal{G}\left(A\{\mathbf{A}_{(\text{author, write, paper})}, \mathbf{A}_{(\text{reader, read, paper})}\}, X\{\mathbf{X}_{(\text{author})}, \mathbf{X}_{(\text{reader})}, \mathbf{X}_{(\text{paper})}\}, Y\{\mathbf{y}_{(\text{paper})}\}\right),$$

AH-UGC proceeds by first partitioning $\mathcal{G}$ by node type and independently applying the coarsening framework to each subgraph. This ensures that only semantically similar nodes are grouped into supernodes and that type-specific structure and features are preserved. Our approach naturally supports varying feature dimensions and allows different coarsening ratios $\eta_{\text{type}}$ across node types. Figure 7 in Appendix H illustrates this process, highlighting how AH-UGC preserves semantic meaning compared to other GC methods that merge heterogeneous nodes indiscriminately.

**Construction of the Coarsened Heterogeneous Graph $\mathcal{G}_c$.** The output of AH-UGC consists of a set of coarsening matrices

$$\mathcal{C}_{\mathcal{H}} = \{\mathcal{C}_{(t)} \in \{0,1\}^{|V_{(t)}| \times |\widetilde{V}_{(t)}|}\}_{t \in \mathcal{T}},$$

each of which maps original nodes of type $t$ i.e., $V_{(t)}$ to their corresponding super-nodes $\widetilde{V}_{(t)}$. Using these mappings, we construct the coarsened graph

$$\mathcal{G}_c\left(\widetilde{A}\{\widetilde{A}_{(\text{author, write, paper})}, \widetilde{A}_{(\text{reader, read, paper})}\}, \widetilde{X}\{\widetilde{X}_{(\text{author})}, \widetilde{X}_{(\text{reader})}, \widetilde{X}_{(\text{paper})}\}, \widetilde{Y}\{\widetilde{y}_{(\text{paper})}\}\right),$$

For each node type $t$, the coarsened feature matrix is computed as: $\widetilde{X}_{(t)} = \mathcal{C}_{(t)} \cdot \mathbf{X}_{(t)}$, where rows of $\mathcal{C}_{(t)}$ are row-normalized so that super-node features represent the average of their constituent nodes. The label matrix $\widetilde{y}_{(\text{paper})}$ is computed by majority voting over the labels of nodes merged into each super-node. To compute the coarsened edge matrices, for each edge type $\mathcal{T}_e \in \mathcal{T}_E$, we consider the interaction between supernodes of types node-type$_1$ and node-type$_2$, corresponding to the edge relation $e = (\text{node-type}_1, \mathcal{T}_e, \text{node-type}_2) \in \widetilde{E}$. The coarsened adjacency matrix $\widetilde{A}_{(e)}$ is then computed as:

$$\widetilde{A}_{(e)} = \mathcal{C}_{(\text{node-type}_1)} \cdot \mathbf{A}_{(e)} \cdot \mathcal{C}^T_{(\text{node-type}_2)}.$$

This formulation accumulates the edge weights between the original nodes to define the inter-supernode connections, thereby preserving the structural connectivity patterns between different node-types of the original graph. Since each edge type is coarsened independently based on the mappings from its corresponding node types, $\mathcal{G}_c$ preserves the heterogeneous semantics and topological relationships of the original graph $\mathcal{G}$. Algorithm 2 in Appendix G outlines the sequence of steps in our AH-UGC framework. By leveraging consistent hashing, our method ensures balanced supernode formation. Theorem 3.2 provides a probabilistic upper bound on the number of nodes mapped to any supernode, thereby guaranteeing load balance across supernodes with high probability.

**Theorem 3.2 (Explicit Load Balance via Random Rightward Merges)** *Let $n$ nodes be sorted according to the consistent hashing scores defined earlier. Let $k$ supernodes be formed by performing $n - k$ random rightward merges in the sorted list. Then, for any constant $c > 0$, the maximum number of nodes in any supernode $S_i$ satisfies:*

$$\Pr\left[\max_i |S_i| \leq \frac{n}{k} + \frac{n(\log k + c)}{k}\right] \geq 1 - e^{-c}$$

**Proof:** The proof is deferred in Appendix C.

## 4 Experiments

We conduct comprehensive experiments to evaluate the effectiveness of AH-UGC. First, we validate its ability to perform *adaptive graph coarsening*. Second, we assess the quality of coarsened graphs using node classification accuracy and spectral similarity. Finally, we demonstrate AH-UGC's generalizability by evaluating its performance on *heterogeneous graphs*.

**Datasets:** We experiment on 23 widely-used benchmark datasets grouped into four categories:

- **Homophilic**: *Cora ,Citeseer, Pubmed* [33], *CS, Physics* [34], *DBLP* [35];
- **Heterophilic**: *Squirrel, Chameleon, Texas, Cornell, Film, Wisconsin* [36–39], *Penn49, deezer-europe, Amherst41, John Hopkins55, Reed98* [11];

- **Heterogeneous**: *IMDB, DBLP, ACM* [7, 25];
- **Large-scale**: *Flickr, Yelp*, [14] *ogbn-arxiv* [6] , *Reddit* [40].

These datasets enable us to evaluate all six key components outlined in Section 2.1. For detailed dataset statistics and characteristics, refer to Table 5 in Appendix A.

**System Specifications:** All experiments are conducted on a server equipped with two **NVIDIA RTX A6000** GPUs (48 GB memory each) and an **Intel Xeon Platinum 8360Y** CPU with **1 TB RAM**.

Table 1: Total time (in seconds) to generate coarsened graphs at multiple resolutions, targeting a set of coarsening ratios of $\mathcal{R} = \{55, 50, 45, 40, 35, 30, 25, 20, 15, 10\}$. The best and the second-best accuracies in each row are highlighted by dark and lighter shades of Green, respectively. "OOT" indicates out-of-time or memory errors.

| Dataset | VAN | VAE | VAC | HE | aJC | aGS | Kron | FGC | LAGC | UGC | AH-UGC |
|---|---|---|---|---|---|---|---|---|---|---|---|
| Cora | 19 | 13 | 29 | 9 | 13 | 30 | 9 | OOT | OOT | 30 | 7 |
| Citeseer | 28 | 23 | 37 | 21 | 22 | 31 | 20 | OOT | OOT | 28 | 6 |
| DBLP | 162 | 138 | 388 | 204 | 206 | 1270 | 184 | OOT | OOT | 131 | 20 |
| PubMed | 166 | 224 | 510 | 213 | 231 | 2351 | 155 | OOT | OOT | 137 | 29 |
| CS | 174 | 237 | 343 | 216 | 256 | 1811 | 204 | OOT | OOT | 233 | 23 |
| Physics | 411 | 798 | 943 | 705 | 906 | 9341 | 755 | OOT | OOT | 331 | 54 |
| Texas | 1.59 | 0.91 | 2.66 | 0.77 | 0.96 | 1.32 | 0.8 | OOT | OOT | 11 | 0.73 |
| Cornell | 1.76 | 0.99 | 2.72 | 0.86 | 1.11 | 1.35 | 0.68 | OOT | OOT | 9 | 0.79 |
| Chameleon | 31 | 17 | 104 | 20 | 32 | 82 | 15 | OOT | OOT | 21 | 6.73 |
| Squirrel | 384 | 61 | 398 | 66 | 342 | 1113 | 68 | OOT | OOT | 53 | 4.69 |
| Film | 64 | 34 | 255 | 36 | 44 | 257 | 30 | OOT | OOT | 92 | 11 |
| Flickr | 1199 | 2301 | 24176 | 2866 | 3421 | 59585 | 2858 | OOT | OOT | 187 | 51 |
| ogbn-arxiv | OOT | OOT | OOT | OOT | OOT | OOT | OOT | OOT | OOT | 1394 | 185 |
| Reddit | OOT | OOT | OOT | OOT | OOT | OOT | OOT | OOT | OOT | 1595 | 290 |
| Yelp | OOT | OOT | OOT | OOT | OOT | OOT | OOT | OOT | OOT | 6904 | 1374 |

## 4.1 Adaptive Coarsening Run-Time.

Given a graph $\mathcal{G}$, we evaluate AH-UGC's ability to adaptively coarsen it to multiple resolutions, targeting a set of coarsening ratios $\mathcal{R} = \{55, 50, 45, 40, 35, 30, 25, 20, 15, 10\}$. As described in Section 3, AH-UGC leverages LSH and consistent hashing to group similar nodes into supernodes, enabling the construction of multiple coarsened graphs in a single pass. This adaptivity significantly reduces computational overhead compared to existing methods, which typically require reprocessing the entire graph for each target resolution. The computational advantages of our approach are evident in Table 1, where AH-UGC outperforms all baseline methods by a significant margin, achieving the lowest coarsening time across all datasets and coarsening ratios, while maintaining scalability even on large-scale graphs where other methods fail.

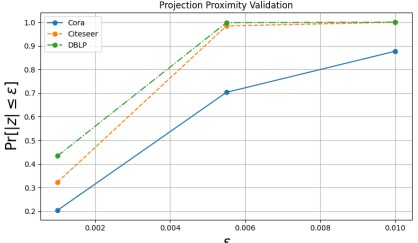

Figure 3: Empirical proof that two feature vectors remain close in projection space.

## 4.2 Spectral Properties Preservation.

Following the experimental setup of [4, 19, 20] we use Hyperbolic Error (HE), Reconstruction Error (RcE) and Relative Eigen Error (REE) to indicate the structural similarity between $\mathcal{G}$ and $\mathcal{G}_c$. A more detailed discussion about these properties is included in Appendix F. Across three spectral evaluation metrics AH-UGC delivers performance that is comparable to, and in several cases surpasses, state-of-the-art methods, see Table 2. While there are minor dips in performance on a few datasets, this trade-off can be justified given the significant computational efficiency and scalability gains offered by our framework. These results underscore that AH-UGC achieves strong structural fidelity without compromising on runtime, making it especially suitable for large-scale or adaptive coarsening scenarios.

**LSH and consistent hashing results.** We empirically validates Theorem 3.1, see Figure 3. As $\epsilon$ increases, $\Pr\left[|h(x) - h(y)| \leq \varepsilon\right]$ approaches 1, consistent with the theoretical erf-based bound. These results justify the use of consistent hashing, where each node is merged with its nearest clockwise neighbor. Theorem 3.1 and Figure 3 together guarantee that similar nodes are projected to nearby locations and are thus highly likely to be merged into a supernode.

Table 2: Illustration of spectral properties preservation, including HE, RcE and REE at 50% coarsening ratio.

| | Dataset | VAN | VAE | VAC | HE | aJC | aGS | Kron | UGC | AH-UGC |
|---|---|---|---|---|---|---|---|---|---|---|
| HE Error | DBLP | 2.20 | 2.07 | 2.21 | 2.21 | 2.12 | 2.06 | 2.24 | 2.10 | 1.99 |
| | Pubmed | 2.49 | 3.33 | 3.46 | 3.19 | 2.77 | 2.48 | 2.74 | 1.72 | 1.53 |
| | Squirrel | 4.17 | 2.61 | 2.72 | 1.52 | 1.92 | 2.01 | 1.87 | 0.69 | 0.82 |
| | Chameleon | 2.77 | 2.55 | 2.99 | 1.80 | 1.86 | 1.97 | 1.86 | 1.28 | 1.71 |
| ReC Error | DBLP | 4.94 | 4.89 | 5.03 | 5.06 | 5.03 | 4.73 | 5.08 | 5.24 | 5.11 |
| | Pubmed | 4.48 | 5.13 | 5.14 | 5.08 | 5.03 | 4.78 | 4.99 | 4.60 | 4.43 |
| | Squirrel | 10.36 | 9.90 | 10.31 | 9.13 | 9.88 | 10.00 | 9.39 | 9.09 | 9.07 |
| | Chameleon | 7.90 | 7.72 | 8.05 | 7.55 | 7.52 | 7.58 | 7.13 | 7.40 | 7.16 |
| REE Error | DBLP | 0.10 | 0.05 | 0.13 | 0.07 | 0.06 | 0.03 | 0.18 | 0.44 | 0.32 |
| | Pubmed | 0.05 | 0.97 | 0.88 | 0.71 | 0.48 | 0.06 | 0.42 | 0.31 | 0.21 |
| | Squirrel | 0.88 | 0.58 | 0.42 | 0.44 | 0.34 | 0.36 | 0.48 | 0.05 | 0.07 |
| | Chameleon | 0.76 | 0.69 | 0.67 | 0.38 | 0.38 | 0.35 | 0.52 | 0.09 | 0.12 |

Table 3: Node classification accuracy across various datasets and models at 50% coarsening ratio.

| Dataset | Model | VAN | VAE | VAC | HE | aJC | aGS | Kron | UGC | AH-UGC | Base |
|---|---|---|---|---|---|---|---|---|---|---|---|
| Citeseer | GCN | 59.90 | 60.36 | 58.40 | 61.26 | 60.81 | 61.26 | 62.76 | 65.31 | 65.46 | 70.12 |
| | SAGE | 66.51 | 65.01 | 64.41 | 63.96 | 66.06 | 65.31 | 63.51 | 61.71 | 64.26 | 74.47 |
| | APPNP | 62.16 | 63.36 | 62.46 | 60.21 | 62.91 | 63.81 | 63.21 | 68.61 | 69.06 | 73.12 |
| PubMed | GCN | 74.34 | 72.46 | 74.06 | 71.72 | 67.36 | 72.87 | 69.59 | 84.66 | 85.47 | 87.60 |
| | SAGE | 74.36 | 73.04 | 73.68 | 66.45 | 69.04 | 74.06 | 71.70 | 87.34 | 72.16 | 88.28 |
| | APPNP | 76.34 | 77.00 | 73.55 | 75.55 | 71.75 | 76.72 | 70.46 | 85.64 | 85.80 | 87.88 |
| Physics | GCN | 94.75 | 94.62 | 94.57 | 94.73 | 94.39 | 94.75 | 94.40 | 95.20 | 94.88 | 95.79 |
| | SAGE | 96.26 | 96.04 | 96.08 | 95.97 | 96.04 | 96.18 | 96.01 | 95.21 | 95.78 | 96.44 |
| | APPNP | 96.20 | 96.20 | 96.28 | 96.11 | 95.97 | 96.07 | 96.21 | 96.17 | 96.10 | 96.28 |
| Chameleon | SGC | 38.60 | 51.58 | 45.79 | 54.91 | 52.63 | 53.15 | 54.39 | 58.60 | 59.65 | 57.46 |
| | Mixhop | 40.53 | 51.40 | 43.33 | 50.35 | 49.82 | 49.30 | 54.39 | 58.25 | 58.60 | 63.16 |
| | GPR-GNN | 40.53 | 46.32 | 41.05 | 39.64 | 40.35 | 43.68 | 51.05 | 54.74 | 52.28 | 55.04 |
| Cornell | SGC | 67.24 | 67.09 | 68.26 | 68.02 | 68.35 | 69.02 | 68.33 | 76.68 | 76.08 | 72.78 |
| | Mixhop | 66.79 | 67.67 | 67.14 | 66.07 | 66.45 | 66.71 | 66.41 | 70.64 | 71.61 | 76.49 |
| | GPR-GNN | 64.98 | 64.27 | 65.17 | 65.00 | 63.55 | 63.67 | 63.48 | 69.66 | 68.00 | 67.46 |
| Penn94 | SGC | 62.93 | 62.33 | 62.23 | 62.13 | 63.52 | 63.03 | 63.52 | 75.74 | 75.87 | 66.78 |
| | Mixhop | 71.71 | 69.62 | 69.35 | 68.36 | 67.98 | 68.40 | 67.98 | 73.36 | 72.13 | 80.28 |
| | GPR-GNN | 68.18 | 68.19 | 68.36 | 68.20 | 67.77 | 68.15 | 68.11 | 67.93 | 68.55 | 79.43 |

## 4.3 Node Classification Accuracy

Graph Neural Networks (GNNs) are widely used for node classification tasks [5, 40–42], where the goal is to predict labels for nodes based on both node features and the underlying graph structure. In this context, we evaluate the effectiveness of AH-UGC by examining how well it preserves predictive performance when downstream models are trained on coarsened graphs [43]. Specifically, we train several GNN models on the coarsened version of the original graph while evaluating their performance on the original graph's test nodes. As discussed earlier, our experimental setup spans a diverse collection of datasets, each with distinct structural characteristics. Following established practice in the literature, we employ different GNN backbones tailored to each graph type. For "homophilic" datasets, we use *GCN* [5], *Sage* [40], *GAT* [41], *GIN* [42] and *APPNP* [43], which are well-suited to leverage dense neighborhood similarity. For "heterophilic" datasets, we adopt *GPRGNN* [44], *MixHop* [45], *H2GNN* [46], *GCN-II* [47], *GatJK* [48] and *SGC* [49], which are designed to handle weak or inverse homophily. For "heterogeneous" graphs, we use *HeteroSGC, HeteroGCN, HeteroGCN2* [25] models that respect node and edge types during message passing. Complete architectural and hyperparameter details are provided in Appendix E. Due to space constraints, Table 3 reports node classification accuracy for homophilic and heterophilic graphs on a representative subset of datasets and GNN models. Please refer to Table 8 in Appendix E for comprehensive results across additional datasets and architectures. The AH-UGC framework consistently delivers results that are either on par with or exceed the performance of existing coarsening methods. As shown in Table 3, the framework is independent of any particular GNN architecture, highlighting its robustness and model-agnostic characteristics.

**Performance on Heterogeneous Graphs:** As outlined in Section 3, conventional graph coarsening techniques struggle with preserving the semantic integrity of heterogeneous graphs. In contrast,

Table 4: Node classification accuracy (%) for heterogeneous datasets at 30% coarsening ratio.

| Dataset | Model | VAN | VAE | VAC | HE | aJC | aGS | Kron | UGC | AH-UGC | Base |
|---|---|---|---|---|---|---|---|---|---|---|---|
| IMDB | HeteroSGC | 27.42 | 27.30 | 27.42 | 27.42 | 27.42 | 27.30 | 27.42 | 50.05 | 57.4 | 66.74 |
| | HeteroGCN | 35.78 | 36.05 | 35.82 | 35.46 | 35.7 | 35.7 | 35.93 | 37.33 | 57.75 | 61.72 |
| | HeteroGCN2 | 35.78 | 35.82 | 35.82 | 35.82 | 35.82 | 35.82 | 35.82 | 37.65 | 58.57 | 63.47 |
| DBLP | HeteroSGC | 30.95 | 29.43 | 29.43 | 53.07 | 56.65 | 29.43 | 29.43 | 37.06 | 79.18 | 94.10 |
| | HeteroGCN | 32.38 | 31.77 | 32.75 | 32.75 | 33 | 35.46 | 31.28 | 63.66 | 66.74 | 84.18 |
| | HeteroGCN2 | 31.69 | 31.52 | 31.77 | 33.25 | 31.12 | 32.01 | 32.63 | 39.08 | 66 | 79.33 |
| ACM | HeteroSGC | 84.46 | 42.31 | OOT | 34.54 | 42.31 | 34.54 | 42.31 | 63.63 | 59 | 92.06 |
| | HeteroGCN | 36.52 | 35.2 | OOT | 35.7 | 35.2 | 35.53 | 35.1 | 38.51 | 84.95 | 92.72 |
| | HeteroGCN2 | 38.67 | 37.35 | OOT | 36.19 | 37.35 | 35.04 | 37.35 | 42.64 | 83.47 | 92.72 |

AH-UGC explicitly enforces type-aware coarsening, ensuring that supernodes are composed of nodes from a single type, thus maintaining the heterogeneity semantics. Table 4 presents node classification accuracies across various heterogeneous GNN models. AH-UGC consistently outperforms other methods due to its ability to preserve type purity within supernodes. This structural consistency enables all tested GNN architectures to achieve significantly higher classification performance. Figure 4 illustrates the degree of supernode impurity for each method. Each bar corresponds to a supernode and depicts the percentage distribution of node types within it. While supernodes generated by AH-UGC are entirely type-pure, those produced by baseline methods exhibit substantial cross-type mixing, leading to semantic drift and reduced model performance. Figure 5 analyzes the effect of increasing coarsening ratios on node classification accuracy. As expected, all methods experience performance degradation with aggressive coarsening. However, the drop is exponential for existing approaches due to rising impurity levels. In contrast, AH-UGC maintains structural purity across coarsening levels, resulting in a gradual, near-linear decline in accuracy. This robustness demonstrates AH-UGC's superior capacity to coarsen heterogeneous graphs while preserving their semantic and structural fidelity.



Figure 4: Supernode impurity across AH-UGC (left), UGC (center) and VAN (right) on IMDB dataset. Different colors represent different node types(*Movie, Director, Actor*).

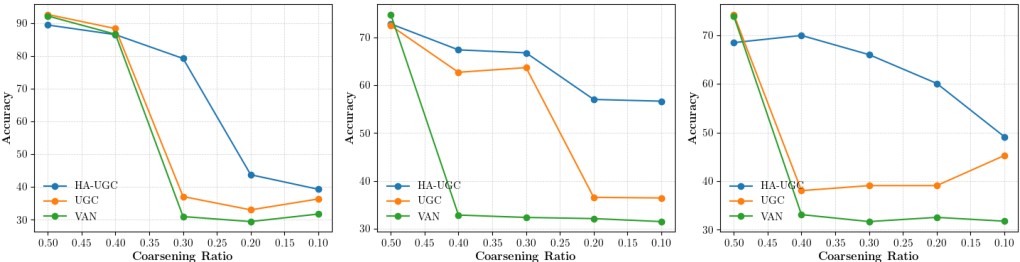

Figure 5: Node classification accuracy on the hDBLP dataset under decreasing coarsening ratios for three heteroGNN models: HeteroSGC (left), HeteroGCN (center), and HeteroGCN2 (right).

## 5 Conclusion

In this paper, we propose AH-UGC, a unified framework for adaptive and heterogeneous graph coarsening. By integrating Locality-Sensitive Hashing (LSH) with Consistent Hashing, AH-UGC efficiently produces multiple coarsened graphs with minimal overhead. Additionally, its type-aware design ensures semantic preservation in heterogeneous graphs by avoiding cross-type node merges. The framework is model-agnostic, scalable, and capable of handling both heterophilic and heterogeneous graphs. We demonstrate that AH-UGC preserves key spectral properties, making it applicable across diverse graph types. Extensive experiments on 23 real-world datasets with various GNN architectures show that AH-UGC consistently outperforms existing methods in scalability, classification accuracy, and structural fidelity.

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

## A  Datasets

We experiment on 24 widely-used benchmark datasets grouped into four categories: **(a) Homophilic**: *Cora ,Citeseer, Pubmed* [33], *CS, Physics* [34], *DBLP* [35]; **(b) Heterophilic**: *Squirrel, Chameleon, Texas, Cornell, Film, Wisconsin* [36–39], *Penn49, deezer-europe, Amherst41, John Hopkins55, Reed98* [11]; **(c) Heterogeneous**: *IMDB, DBLP, ACM* [7, 25]; and **(d) Large-scale**: *Flickr, Yelp,* [14] *ogbn-arxiv* [6] , *Reddit* [40]. These datasets enable us to evaluate all six key components outlined in Section 2.1. Please refer to Table 5 and  6 for detailed dataset statistics and characteristics.

Table 5: Summary of the datasets.

| Category | Data | Nodes | Edges | Feat. | Class | H.R($\alpha$) |
|---|---|---|---|---|---|---|
| Homophilic dataset | Cora | 2,708 | 5,429 | 1,433 | 7 | 0.19 |
| | Citeseer | 3,327 | 9,104 | 3,703 | 6 | 0.26 |
| | DBLP | 17,716 | 52,867 | 1,639 | 4 | 0.18 |
| | CS | 18,333 | 163,788 | 6,805 | 15 | 0.20 |
| | PubMed | 19,717 | 44,338 | 500 | 3 | 0.20 |
| | Physics | 34,493 | 247,962 | 8,415 | 5 | 0.07 |
| Heterophilic dataset | Texas | 183 | 309 | 1703 | 5 | 0.91 |
| | Cornell | 183 | 295 | 1703 | 5 | 0.70 |
| | Film | 7600 | 33544 | 931 | 5 | 0.78 |
| | Squirrel | 5201 | 217073 | 2089 | 5 | 0.78 |
| | Chameleon | 2277 | 36101 | 2325 | 5 | 0.75 |
| | Penn94 | 41,554 | 1.36M | 5 | 2 | 0.53 |
| | Deezer-europe | 28,281 | 185.5k | 31.24k | 2 | - |
| | Amherst41 | 2235 | 181.9k | 1193 | 3 | - |
| | John-Hopkin55 | 41,554 | 2.7M | 4,814 | 3 | - |
| | Reed98 | 962 | 37.6k | 745 | 3 | - |
| Large dataset | Flickr | 89,250 | 899,756 | 500 | 7 | - |
| | Reddit | 232,965 | 11.60M | 602 | 41 | - |
| | Ogbn-arxiv | 169,343 | 1.16M | 128 | 40 | - |
| | Yelp | 716,847 | 13.95M | 300 | 100 | - |

Table 6: Summary of Heterogeneous graph datasets

| Dataset | Nodes | Edges | Features | Classes |
|---|---|---|---|---|
| IMDB | Movie - 4278
Director - 2081
Actor - 5257 | (Movie, to, Director) - 4278
(Movie, to, Actor) - 12828
(Director, to, Movie) - 4278
(Actor, to, Movie) - 12828 | 3061 | Movie: 3 |
| DBLP | Author - 4057
Paper - 4231
Term - 7723
Conference - 50 | (Author, to, Paper) - 19645
(Paper, to, Author) - 19645
(Paper, to, Term) - 85810
(Paper, to, Conference) - 14328
(Term, to, Paper) - 85810
(Conference, to, Paper) - 14328 | Author - 334
Paper - 4231
Term - 50
Conference - NA | Author: 4 |
| ACM | Paper - 3025
Author - 5959
Subject - 56
Term - 1902 | (Paper, cite, Paper) - 5343
(Paper, ref, Paper) - 5343
(Paper, to, Author) - 9949
(Author, to, Paper) - 9949
(Paper, to, Subject) - 3025
(Subject, to, Paper) - 3025
(Paper, to, Term) - 255619
(Term, to, Paper) - 255619 | All except term - 1902
Term - NA | Paper: 3 |

## B  Locality-Sensitive Hashing and Consistent Hashing

Locality-Sensitive Hashing (LSH) is a technique for hashing high-dimensional data points so that similar items are more likely to collide (i.e., hash to the same bucket) [32, 50, 51]. It is commonly used in approximate nearest neighbor search, dimensionality reduction, and randomized algorithms [52]. For example, a hash function $h(\cdot)$ is locality-sensitive with respect to a similarity measure $s(\cdot, \cdot)$ if $\Pr[h(x) = h(y)]$ increases with $s(x, y)$. Gaussian LSH schemes, such as those using random projections, are particularly effective for preserving Euclidean distances [4, 22].

In the consistent hashing (CH) [23, 24] scheme, objects/requests are hashed to random bins/servers on the unit circle, as shown in Figure 6. Objects are then assigned to the closest bin in the clockwise direction. CH was originally proposed for load balancing in distributed systems; it maps data points

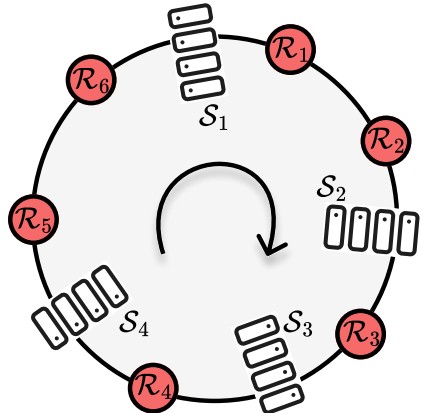

Figure 6: Consistent Hashing (CH): Objects and bins are hashed to a unit circle; each object is assigned to the next bin in clockwise order.

to buckets such that small changes in input (e.g., adding or removing an object) do not drastically affect the overall assignment. We aim to employ CH for adaptive graph coarsening, as it enables stable and scalable grouping of similar objects/nodes. When combined with LSH, consistent hashing offers a powerful mechanism for adaptive graph reduction.

## C  Proof of Theorem 3.2

**Theorem C.1 (Explicit Load Balance via Random Rightward Merges)** *Let $n$ nodes be sorted according to the consistent hashing scores defined earlier. Let $k$ supernodes be formed by performing $n - k$ random rightward merges in the sorted list. Then, for any constant $c > 0$, the maximum number of nodes in any supernode $S_i$ satisfies:*

$$\Pr\left[\max_i |S_i| \leq \frac{n}{k} + \frac{n(\log k + c)}{k}\right] \geq 1 - e^{-c}$$

**Proof** Let $U_1, \ldots, U_{k-1} \sim \text{Uniform}(0, 1)$ and let $U_{(1)} < \cdots < U_{(k-1)}$ be their order statistics. Define the spacings:

$$I_1 = U_{(1)} - 0, \quad I_2 = U_{(2)} - U_{(1)}, \quad \ldots, \quad I_k = 1 - U_{(k-1)}$$

Then $(I_1, \ldots, I_k)$ form a random partition of the unit interval $[0, 1]$. It is a classical result (e.g., [53]) that:

- The vector $(I_1, \ldots, I_k) \sim \text{Dirichlet}(1, \ldots, 1)$,
- Each individual spacing $I_i \sim \text{Beta}(1, k - 1)$.

**Tail bound on $I_i$.** The PDF of $I_i$ is:

$$f(t) = (k - 1)(1 - t)^{k-2}, \quad t \in [0, 1]$$

and its tail probability is:

$$\Pr[I_i > t] = (1 - t)^{k-1}$$

Choose $t = \frac{\log k + c}{k}$. Then:

$$\Pr[I_i > t] \leq \exp\left(-(\log k + c)\right) = \frac{1}{k} e^{-c}$$

**Union bound.** Over all $k$ intervals:

$$\Pr\left[\max_i I_i > \frac{\log k + c}{k}\right] \leq k \cdot \frac{1}{k} e^{-c} = e^{-c} \Rightarrow \Pr\left[\max_i I_i \leq \frac{\log k + c}{k}\right] \geq 1 - e^{-c}$$

**Scaling to $n$ nodes.** We model the sorted list of $n$ nodes as uniformly spaced over $[0, 1]$. Each spacing $I_i$ then corresponds to a fraction of the list, and multiplying by $n$ yields the expected number

of nodes in that supernode:

$$|S_i| = n \cdot I_i \Rightarrow \max_i |S_i| = n \cdot \max_i I_i \leq \frac{n}{k} + \frac{n(\log k + c)}{k}$$

This completes the proof.

# D   Proof of Theorem 3.1

**Theorem D.1 (Projection Proximity for Similar Points)** *Let $x, y \in \mathbb{R}^d$, and define the projection function:*

$$h(x) = \sum_{j=1}^{\ell} r_j^\top x, \quad r_j \sim \mathcal{N}(0, I_d) \text{ i.i.d.}$$

*Then the difference $h(x) - h(y) \sim \mathcal{N}(0, \ell \|x - y\|^2)$, and for any $\varepsilon > 0$:*

$$\Pr\left[|h(x) - h(y)| \leq \varepsilon\right] = \text{erf}\left(\frac{\varepsilon}{\sqrt{2\ell}\|x - y\|}\right)$$

**Proof** Let $z = x - y \in \mathbb{R}^d$. Then:

$$h(x) - h(y) = \sum_{j=1}^{\ell} r_j^\top x - \sum_{j=1}^{\ell} r_j^\top y = \sum_{j=1}^{\ell} r_j^\top (x - y) = \sum_{j=1}^{\ell} r_j^\top z$$

Each term $r_j^\top z$ is a linear projection of a standard Gaussian vector, hence:

$$r_j^\top z \sim \mathcal{N}(0, \|z\|^2) = \mathcal{N}(0, \|x - y\|^2)$$

Since the $r_j$ are independent, the sum of $\ell$ such independent variables is:

$$h(x) - h(y) \sim \mathcal{N}(0, \ell \|x - y\|^2)$$

Now consider the probability:
$$\Pr\left[|h(x) - h(y)| \leq \varepsilon\right]$$

This is the cumulative probability within $\varepsilon$ of a zero-mean Gaussian with variance $\ell \|x - y\|^2$. Let $\sigma^2 = \ell \|x - y\|^2$. Then:

$$\Pr\left[|Z| \leq \varepsilon\right] = \text{erf}\left(\frac{\varepsilon}{\sqrt{2\sigma^2}}\right) = \text{erf}\left(\frac{\varepsilon}{\sqrt{2\ell}\|x - y\|}\right)$$

as required.

# E   Node Classification Accuracy

Graph Neural Networks (GNNs), designed to operate on graph data [4, 54], have demonstrated strong performance across a range of applications [55–58]. Nevertheless, their scalability to large graphs remains a significant bottleneck. Motivated by recent efforts in scalable learning [43], we explore how our graph coarsening framework can improve the efficiency and scalability of GNN training, enabling more effective processing of large-scale graph data. Specifically, we train several GNN models on the coarsened version of the original graph while evaluating their performance on the original graph's test nodes. As discussed earlier in 4.3, our experimental setup spans a diverse collection of datasets, each with distinct structural characteristics. For *homophilic* graph settings, we follow the architectural configurations proposed in UGC [4], see Table 7. For *heterophilic* graphs, the GNN model designs are based on the implementations introduced in [11]. The *heterogeneous* GNN architectures are adopted directly from [25].

Table 8 reports node classification accuracy for homophilic and Table 9 reports node classification accuracy for heterophilic graphs. The AH-UGC framework consistently delivers results that are either on par with or exceed the performance of existing coarsening methods. As shown in Table 3, the framework is independent of any particular GNN architecture, highlighting its robustness and model-agnostic characteristics.

Table 7: Summary of GNN architectures used in our experiments. Each model is described by its layer composition, hidden units, activation functions, dropout strategy, and notable characteristics.

| Model | Layers | Hidden Units | Activation | Dropout | Learning rate | Decay | Epoch |
|---|---|---|---|---|---|---|---|
| GCN | 3 × GCNConv | 64 → 64 → Output | ReLU | Yes (intermediate layers) | 0.003 | 0.0005 | 500 |
| APPNP | Linear → Linear → APPNP | 64 → 64 → 10 → Output | ReLU | Yes (before Linear layers) | 0.003 | 0.0005 | 500 |
| GAT | 2 × GATv2Conv | 64 × 8 → Output | ELU | Yes (p=0.6) | 0.003 | 0.0005 | 500 |
| GIN | 2 × GATv2Conv | 64 × 8 → Output | ELU | Yes (p=0.6) | 0.003 | 0.0005 | 500 |
| GraphSAGE | 2 × SAGEConv | 64 → Output | ReLU | Yes (after first layer) | 0.003 | 0.0005 | 500 |

Table 8: Node classification accuracy (%) for homophilic datasets.

| Dataset | Model | VAN | VAE | VAC | HE | aJC | aGS | Kron | UGC | AH-UGC | Base |
|---|---|---|---|---|---|---|---|---|---|---|---|
| Cora | GCN | 77.34 | 83.79 | 81.58 | 81.58 | 83.05 | 82.32 | 79.18 | 79.00 | 77.34 | 85.81 |
| | SAGE | 80.47 | 82.87 | 81.95 | 81.76 | 83.97 | 82.87 | 82.87 | 76.61 | 76.24 | 89.87 |
| | GIN | 78.63 | 77.53 | 74.58 | 76.79 | 79.18 | 78.08 | 77.16 | 55.43 | 77.34 | 87.29 |
| | GAT | 77.16 | 78.08 | 75.87 | 74.40 | 81.21 | 80.47 | 74.58 | 78.26 | 81.03 | 87.10 |
| | APPNP | 82.87 | 84.53 | 82.50 | 84.53 | 84.34 | 85.26 | 82.87 | 86.37 | 84.53 | 88.58 |
| DBLP | GCN | 79.65 | 80.36 | 80.55 | 79.99 | 80.55 | 79.26 | 79.40 | 85.75 | 80.27 | 84.00 |
| | SAGE | 80.58 | 80.07 | 80.16 | 80.81 | 80.61 | 81.57 | 79.48 | 68.56 | 68.31 | 84.08 |
| | GIN | 79.40 | 79.20 | 80.38 | 78.83 | 77.96 | 78.18 | 78.01 | 73.95 | 79.82 | 83.26 |
| | GAT | 74.43 | 78.32 | 76.49 | 77.56 | 78.97 | 77.51 | 75.93 | 77.93 | 79.48 | 82.25 |
| | APPNP | 84.25 | 83.80 | 83.63 | 83.60 | 83.29 | 84.25 | 84.05 | 84.84 | 85.18 | 85.75 |
| CS | GCN | 91.63 | 92.01 | 91.19 | 92.03 | 91.41 | 87.26 | 92.55 | 92.66 | 92.47 | 93.51 |
| | SAGE | 94.32 | 94.19 | 94.57 | 94.24 | 93.94 | 93.70 | 94.02 | 89.17 | 89.83 | 94.82 |
| | GIN | 89.80 | 89.69 | 89.83 | 90.70 | 89.61 | 88.00 | 90.64 | 86.77 | 81.07 | 83.50 |
| | GAT | 91.98 | 91.52 | 92.31 | 91.57 | 90.67 | 91.19 | 89.50 | 89.83 | 90.48 | 91.84 |
| Citeseer | GCN | 59.90 | 60.36 | 58.40 | 61.26 | 60.81 | 61.26 | 62.76 | 65.31 | 65.46 | 70.12 |
| | SAGE | 66.51 | 65.01 | 64.41 | 63.96 | 66.06 | 65.31 | 63.51 | 61.71 | 64.26 | 74.47 |
| | GIN | 59.60 | 60.36 | 59.00 | 59.45 | 56.15 | 62.91 | 57.50 | 64.41 | 63.66 | 71.62 |
| | GAT | 53.45 | 58.55 | 54.95 | 53.45 | 62.76 | 59.75 | 57.35 | 65.76 | 69.21 | 71.32 |
| | APPNP | 62.16 | 63.36 | 62.46 | 60.21 | 62.91 | 63.81 | 63.21 | 68.61 | 69.06 | 73.12 |
| PubMed | GCN | 74.34 | 72.46 | 74.06 | 71.72 | 67.36 | 72.87 | 69.59 | 84.66 | 85.47 | 87.60 |
| | SAGE | 74.36 | 73.04 | 73.68 | 66.45 | 69.04 | 74.06 | 71.70 | 87.34 | 72.16 | 88.28 |
| | GIN | 57.17 | 66.53 | 61.53 | 60.11 | 65.66 | 60.85 | 63.46 | 82.42 | 83.97 | 85.75 |
| | GAT | 46.85 | 40.03 | 52.68 | 50.60 | 53.29 | 56.99 | 69.09 | 84.66 | 84.63 | 87.39 |
| | APPNP | 76.34 | 77.00 | 73.55 | 75.55 | 71.75 | 76.72 | 70.46 | 85.64 | 85.80 | 87.88 |
| Physics | GCN | 94.75 | 94.62 | 94.57 | 94.73 | 94.39 | 94.75 | 94.40 | 95.20 | 94.88 | 95.79 |
| | SAGE | 96.26 | 96.04 | 96.08 | 95.97 | 96.04 | 96.18 | 96.01 | 95.21 | 95.78 | 96.44 |
| | GIN | 94.90 | 94.56 | 94.78 | 94.49 | 93.79 | 94.79 | 92.65 | 94.41 | 94.94 | 95.66 |
| | GAT | 94.97 | 95.01 | 95.00 | 94.65 | 95.36 | 94.60 | 94.85 | 96.02 | 95.10 | 94.28 |
| | APPNP | 96.20 | 96.20 | 96.28 | 96.11 | 95.97 | 96.07 | 96.21 | 96.17 | 96.10 | 96.28 |

## F  Spectral Properties

1. **Relative Eigen Error (REE):** REE used in [4, 19, 20] gives the means to quantify the measure of the eigen properties of the original graph $\mathcal{G}$ that are preserved in coarsened graph $\mathcal{G}_c$.

   **Definition F.1** *REE is defined as follows:*

$$REE(L, L_c, k) = \frac{1}{k} \sum_{i=1}^{k} \frac{|\widetilde{\lambda}_i - \lambda_i|}{\lambda_i} \tag{1}$$

   *where $\lambda_i$ and $\widetilde{\lambda}_i$ are top $k$ eigenvalues of original graph Laplacian (L) and coarsened graph Laplacian ($L_c$) matrix, respectively.*

2. **Hyperbolic error (HE):** HE [59] indicates the structural similarity between $\mathcal{G}$ and $\mathcal{G}_c$ with the help of a lifted matrix along with the feature matrix $X$ of the original graph.

   **Definition F.2** *HE is defined as follows:*

$$HE = arccosh\left(\frac{||(L - L_{\text{lift}})X||_F^2 ||X||_F^2}{2 trace(X^T L X) trace(X^T L_{\text{lift}} X)} + 1\right) \tag{2}$$

Table 9: Node classification accuracy (%) for heterophilic datasets.

| Dataset | Model | VAN | VAE | VAC | HE | aJC | aGS | Kron | UGC | AH-UGC | Base |
|---|---|---|---|---|---|---|---|---|---|---|---|
| Film | SGC | 29.36 | 27.84 | 29.95 | 26.15 | 26.89 | 25.74 | 27.74 | 21.47 | 21.68 | 27.63 |
| | Mixhop | 28.21 | 30.68 | 29.84 | 29.52 | 29.10 | 29.15 | 31.15 | 21.57 | 21.79 | 30.92 |
| | GCN2 | 26.15 | 28.47 | 28.00 | 26.94 | 27.63 | 25.84 | 29.42 | 19.47 | 20.42 | 28.36 |
| | GPR-GNN | 26.52 | 27.95 | 27.10 | 27.74 | 26.78 | 28.36 | 28.26 | 20.68 | 21.31 | 29.73 |
| | GatJK | 26.11 | 25.89 | 25.79 | 25.10 | 25.31 | 25.31 | 26.63 | 22.42 | 21.21 | 23.94 |
| deezer-europe | SGC | 54.55 | 55.31 | 54.50 | 55.38 | 54.48 | 54.69 | 55.15 | 54.49 | 55.06 | 57.08 |
| | Mixhop | 58.42 | 59.10 | 58.48 | 58.82 | 58.34 | 57.38 | 58.80 | 59.78 | 60.98 | 64.31 |
| | GCN2 | 57.79 | 58.34 | 57.76 | 58.34 | 57.15 | 57.57 | 58.25 | 58.00 | 58.46 | 60.88 |
| | GPR-GNN | 56.30 | 56.85 | 56.70 | 56.77 | 55.73 | 55.55 | 56.31 | 58.44 | 58.46 | 56.97 |
| | GatJK | 55.21 | 57.50 | 54.63 | 55.76 | 55.31 | 56.03 | 56.87 | 57.01 | 57.33 | 59.01 |
| Amherst41 | SGC | 61.42 | 63.19 | 59.06 | 60.83 | 63.39 | 62.99 | 63.78 | 78.74 | 73.82 | 73.46 |
| | Mixhop | 59.25 | 58.46 | 57.68 | 58.66 | 59.06 | 63.78 | 58.66 | 69.29 | 64.37 | 72.48 |
| | GCN2 | 62.99 | 62.01 | 60.63 | 59.25 | 58.66 | 60.63 | 56.50 | 71.06 | 68.50 | 71.74 |
| | GPR-GNN | 59.45 | 58.86 | 58.07 | 55.91 | 57.68 | 59.25 | 55.71 | 66.73 | 63.98 | 60.93 |
| | GatJK | 57.48 | 63.58 | 60.24 | 62.99 | 61.61 | 64.76 | 62.60 | 64.37 | 67.72 | 78.13 |
| Johns Hopkins55 | SGC | 62.72 | 69.19 | 68.77 | 69.35 | 68.85 | 70.28 | 69.19 | 73.80 | 72.96 | 73.77 |
| | Mixhop | 63.64 | 65.74 | 68.18 | 64.90 | 62.22 | 64.90 | 63.73 | 69.94 | 67.25 | 73.56 |
| | GCN2 | 66.16 | 67.51 | 67.42 | 64.23 | 65.49 | 65.74 | 64.40 | 71.12 | 65.24 | 73.45 |
| | GPR-GNN | 62.05 | 63.06 | 62.30 | 62.80 | 60.37 | 61.96 | 61.71 | 66.33 | 63.31 | 64.95 |
| | GatJK | 62.80 | 69.10 | 67.34 | 66.41 | 65.99 | 65.58 | 67.00 | 69.77 | 65.32 | 77.12 |
| Reed98 | SGC | 53.46 | 57.14 | 53.92 | 52.07 | 55.30 | 58.06 | 53.92 | 57.60 | 57.60 | 68.79 |
| | Mixhop | 50.69 | 58.99 | 49.77 | 48.85 | 55.30 | 59.45 | 53.46 | 60.37 | 52.53 | 62.43 |
| | GCN2 | 56.68 | 59.45 | 51.61 | 50.69 | 51.61 | 56.68 | 50.69 | 61.75 | 57.14 | 64.16 |
| | GPR-GNN | 48.39 | 57.60 | 48.39 | 45.62 | 55.76 | 58.06 | 53.46 | 57.60 | 54.84 | 56.07 |
| | GatJK | 55.30 | 58.99 | 53.00 | 51.61 | 51.61 | 56.22 | 53.92 | 62.67 | 60.83 | 69.94 |
| Squirrel | SGC | 31.97 | 33.13 | 30.98 | 36.66 | 34.97 | 36.59 | 35.59 | 40.89 | 39.51 | 43.61 |
| | Mixhop | 36.28 | 30.21 | 24.60 | 34.90 | 28.44 | 27.90 | 37.05 | 46.12 | 43.97 | 46.40 |
| | GCN2 | 39.74 | 42.28 | 39.20 | 41.74 | 37.97 | 39.12 | 41.51 | 43.12 | 44.35 | 50.72 |
| | GPR-GNN | 29.36 | 25.67 | 28.82 | 28.82 | 26.44 | 27.06 | 30.59 | 45.12 | 43.74 | 34.39 |
| | GatJK | 31.44 | 37.43 | 32.82 | 46.12 | 38.36 | 37.89 | 46.81 | 40.89 | 39.43 | 46.01 |
| Chameleon | SGC | 38.60 | 51.58 | 45.79 | 54.91 | 52.63 | 53.15 | 54.39 | 58.60 | 59.65 | 57.46 |
| | Mixhop | 40.53 | 51.40 | 43.33 | 50.35 | 49.82 | 49.30 | 54.39 | 58.25 | 58.60 | 63.16 |
| | GCN2 | 47.37 | 52.11 | 56.84 | 59.30 | 59.65 | 58.95 | 59.12 | 51.40 | 49.82 | 67.11 |
| | GPR-GNN | 40.53 | 46.32 | 41.05 | 39.64 | 40.35 | 43.68 | 51.05 | 54.74 | 52.28 | 55.04 |
| | GatJK | 41.40 | 52.46 | 36.49 | 60.00 | 56.49 | 55.96 | 62.63 | 54.39 | 55.44 | 71.05 |
| Cornell | SGC | 67.24 | 67.09 | 68.26 | 68.02 | 68.35 | 69.02 | 68.33 | 76.68 | 76.08 | 72.78 |
| | Mixhop | 66.79 | 67.67 | 67.14 | 66.07 | 66.45 | 66.71 | 66.41 | 70.64 | 71.61 | 76.49 |
| | GCN2 | 66.31 | 66.83 | 66.98 | 67.64 | 67.17 | 66.50 | 66.50 | 72.71 | 70.90 | 77.18 |
| | GPR-GNN | 64.98 | 64.27 | 65.17 | 65.00 | 63.55 | 63.67 | 63.48 | 69.66 | 68.00 | 67.46 |
| | GatJK | 63.48 | 65.31 | 68.28 | 66.00 | 67.40 | 66.21 | 66.64 | 70.09 | 70.35 | 78.37 |
| Penn94 | SGC | 62.93 | 62.33 | 62.23 | 62.13 | 63.52 | 63.03 | 63.52 | 75.74 | 75.87 | 66.78 |
| | Mixhop | 71.71 | 69.62 | 69.35 | 68.36 | 67.98 | 68.40 | 67.98 | 73.36 | 72.13 | 80.28 |
| | GCN2 | 71.79 | 69.55 | 70.75 | 69.52 | 69.61 | 71.41 | 69.61 | 71.85 | 72.07 | 81.75 |
| | GPR-GNN | 68.18 | 68.19 | 68.36 | 68.20 | 67.77 | 68.15 | 68.11 | 67.93 | 68.55 | 79.43 |
| | GatJK | 67.94 | 67.05 | 66.73 | 66.21 | 66.34 | 66.06 | 66.33 | 69.23 | 69.26 | 80.74 |

where $L$ is the Laplacian matrix and $X \in \mathbb{R}^{N \times d}$ is the feature matrix of the original input graph, $L_{\text{lift}}$ is the lifted Laplacian matrix defined in [20] as $L_{\text{lift}} = \mathcal{C} L_c \mathcal{C}^T$ where $\mathcal{C} \in \mathbb{R}^{N \times n}$ is the coarsening matrix and $L_c$ is the Laplacian of $\mathcal{G}_c$.

3. **Reconstruction Error (RcE)**

**Definition F.3** *Let $L$ be the original Laplacian matrix and $L_{lift}$ be the lifted Laplacian matrix, then the reconstruction error (RE) [19, 60] is defined as:*

$$RcE = \|L - L_{lift}\|_F^2 \tag{3}$$

# G   Algorithms

# H   Heterogenous graph coarsening

Table 10: This table illustrates spectral properties including HE, RcE, REE across datasets and methods at 50% coarsening ratio. AH-UGC achieves competitive performance across most datasets.

| | Dataset | VAN | VAE | VAC | HE | aJC | aGS | Kron | UGC | AH-UGC |
|---|---|---|---|---|---|---|---|---|---|---|
| **HE Error** | Cora | 2.04 | 2.08 | 2.14 | 2.19 | 2.13 | 1.95 | 2.14 | 1.96 | 2.03 |
| | DBLP | 2.20 | 2.07 | 2.21 | 2.21 | 2.12 | 2.06 | 2.24 | 2.10 | 1.99 |
| | Pubmed | 2.49 | 3.33 | 3.46 | 3.19 | 2.77 | 2.48 | 2.74 | 1.72 | 1.53 |
| | Squirrel | 4.17 | 2.61 | 2.72 | 1.52 | 1.92 | 2.01 | 1.87 | 0.69 | 0.82 |
| | Chameleon | 2.77 | 2.55 | 2.99 | 1.80 | 1.86 | 1.97 | 1.86 | 1.28 | 1.71 |
| | Deezer-Europe | 1.90 | 1.97 | 2.04 | 1.95 | 1.90 | 1.62 | 1.90 | 1.76 | 1.61 |
| | Penn94 | 1.96 | 1.52 | 1.65 | 1.57 | 1.51 | 1.43 | 1.55 | 1.05 | 1.09 |
| **ReC Error** | Cora | 3.78 | 3.83 | 3.90 | 3.95 | 3.91 | 3.71 | 3.92 | 4.07 | 4.14 |
| | DBLP | 4.94 | 4.89 | 5.03 | 5.06 | 5.03 | 4.73 | 5.08 | 5.24 | 5.11 |
| | Pubmed | 4.48 | 5.13 | 5.14 | 5.08 | 5.03 | 4.78 | 4.99 | 4.60 | 4.43 |
| | Squirrel | 10.36 | 9.90 | 10.31 | 9.13 | 9.88 | 10.00 | 9.39 | 9.09 | 9.07 |
| | Chameleon | 7.90 | 7.72 | 8.05 | 7.55 | 7.52 | 7.58 | 7.13 | 7.40 | 7.16 |
| | Deezer-Europe | 5.08 | 5.06 | 5.19 | 5.04 | 5.04 | 4.68 | 5.01 | 8.03 | 8.05 |
| | Penn94 | 7.77 | 7.71 | 7.77 | 7.73 | 7.73 | 7.63 | 7.76 | 7.71 | 7.74 |
| **REE Error** | Cora | 0.09 | 0.07 | 0.05 | 0.04 | 0.11 | 0.09 | 0.03 | 0.64 | 0.66 |
| | DBLP | 0.10 | 0.05 | 0.13 | 0.07 | 0.06 | 0.03 | 0.18 | 0.44 | 0.32 |
| | Pubmed | 0.05 | 0.97 | 0.88 | 0.71 | 0.48 | 0.06 | 0.42 | 0.31 | 0.21 |
| | Squirrel | 0.88 | 0.58 | 0.42 | 0.44 | 0.34 | 0.36 | 0.48 | 0.05 | 0.07 |
| | Chameleon | 0.76 | 0.69 | 0.67 | 0.38 | 0.38 | 0.35 | 0.52 | 0.09 | 0.12 |
| | Deezer-Europe | 0.48 | 0.29 | 0.47 | 0.25 | 0.21 | 0.02 | 0.19 | 0.35 | 0.35 |
| | Penn94 | 0.31 | 0.02 | 0.05 | 0.02 | 0.09 | 0.05 | 0.08 | 0.22 | 0.23 |

---

**Algorithm 1** AH-UGC: Adaptive Universal Graph Coarsening

---

**Require:** Input $\mathcal{G}(V, A, X)$, $l \leftarrow$ Number of Projectors
1: $\alpha = \frac{|\{(v,u) \in E : y_v = y_u\}|}{|E|}$; $\alpha$ is heterophily factor, $y_i \in \mathbb{R}^N$ is node labels, $E$ denotes edge list
2: $F = \{(1 - \alpha) \cdot X \oplus \alpha \cdot A\}$
3: $\mathcal{S} \leftarrow F \cdot \mathcal{W} + b; \mathcal{S} \in \mathbb{R}^{n \times l}$          // compute projections
4: $\mathcal{W} \in \mathbb{R}^{d \times l}, \; b \in \mathbb{R}^l \sim \mathcal{D}(\cdot)$          // sample projections
5: $\mathcal{S} \leftarrow F \cdot \mathcal{W} + b; \; \mathcal{S} \in \mathbb{R}^{n \times l}$          // compute projections
6: $s_i \leftarrow \text{AGGREGATE}(\{\mathcal{S}_{i,k}\}_{k=1}^l) = \frac{1}{l} \sum_{k=1}^l \mathcal{S}_{i,k} \quad \forall i \in \{1, \ldots, n\}$    // mean aggregation
7: $\mathcal{L} \leftarrow \text{sort}(\{v_i\}_{i=1}^n)$ by ascending $s_i$          // ordered node list
8: $\mathcal{L} \leftarrow [\{u_1 : \{v_1\}\}, \{u_2 : \{v_2\}\}, \ldots, \{u_n : \{v_n\}\}]$      // initial super-nodes
9: **while** $|\mathcal{L}|/|V| > r$ **do**
10:     $u_j \sim \text{Uniform}(\mathcal{L})$          // sample a super-node
11:     $\mathcal{L}[u_j] \leftarrow \mathcal{L}[u_j] \cup \mathcal{L}[u_{j+1}]$          // merge with right neighbor
12:     $\mathcal{L} \leftarrow \mathcal{L} \setminus \{u_{j+1}\}$          // remove right neighbor
13: $\mathcal{C} \in \{0, 1\}^{|\mathcal{L}| \times |V|}, \; \mathcal{C}_{ij} \leftarrow \begin{cases} 1 & \text{if } v_j \in \mathcal{L}[u_i] \\ 0 & \text{otherwise} \end{cases}$      // partition matrix
14: $\mathcal{C} \leftarrow \text{row-normalize}(\mathcal{C})$          // normalize rows: $\sum_j \mathcal{C}_{ij} = 1$
15: $\widetilde{F} \leftarrow \mathcal{C}F \quad ; \quad \widetilde{A} \leftarrow \mathcal{C}A\mathcal{C}^T$          // coarsened features and adjacency
16: **return** $\mathcal{G}_c = (\widetilde{V}, \widetilde{A}, \widetilde{F}), \; \mathcal{C}$

---

**Algorithm 2** Heterogeneous Graph Coarsening

**Require:** Graph $\mathcal{G}\left(\{X_{(\text{node\_type})}\}, \{A_{(\text{edge\_type})}\}, \{y_{(\text{target\_type})}\}\right)$, compression ratio $\eta$

**Ensure:** Condensed graph $\mathcal{G}_c\left(\{\widetilde{X}_{(\text{node\_type})}\}, \{\widetilde{A}_{(\text{edge\_type})}\}, \{\widetilde{Y}_{(\text{target\_type})}\}\right)$

1: **for** each node type $t$ **do**
2:     $r_t \leftarrow \eta \cdot |V_t|$
3:     $\mathcal{G}_t^{\text{coarse}}, \mathcal{C}_t \leftarrow \text{AH-UGC}(X_t, A_t, r_t)$
4:     $\widetilde{X}_t \leftarrow$ node features from $\mathcal{G}_t^{\text{coarse}}$
5:     **if** $t$ is target type **then**
6:         $\widetilde{y}_t[i] \leftarrow$ majority vote of $y_j$ for $v_j \in \mathcal{C}_t[i]$
7: **for** each edge type $e = (t_1, t_2)$ **do**
8:     Initialize $\widetilde{A}_e \in \mathbb{R}^{|\widetilde{V}_{t_1}| \times |\widetilde{V}_{t_2}|}$
9:     **for** each $(v_i, v_j) \in A_e$ **do**
10:         $u \leftarrow$ super-node index of $v_i$ via $\mathcal{C}_{t_1}$
11:         $v \leftarrow$ super-node index of $v_j$ via $\mathcal{C}_{t_2}$
12:         $\widetilde{A}_e[u, v] \leftarrow \widetilde{A}_e[u, v] + 1$
13: **return** $\mathcal{G}_c\left(\{\widetilde{X}_{(\text{node\_type})}\}, \{\widetilde{A}_{(\text{edge\_type})}\}, \{\widetilde{Y}_{(\text{target\_type})}\}\right)$

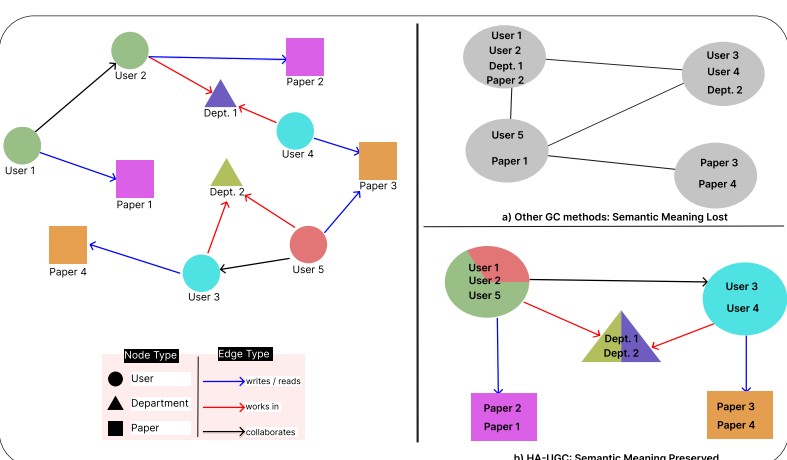

Figure 7: This figure illustrates this process, highlighting how AH-UGC preserves semantic meaning compared to other GC methods that merge heterogeneous nodes indiscriminately.

