# OpenReview forum: "AH-UGC: $\underline{\text{A}}$daptive and $\underline{\text{H}}$eterogeneous-$\underline{\text{U}}$niversal $\underline{\text{G}}$raph $\underline{\text{C}}$oarsening"
_NeurIPS.cc/2025/Conference — Submitted to NeurIPS 2025_

### Official Review · Reviewer_rETW · 2025-06-24

**Clarity:** 2
**Significance:** 2
**Originality:** 2
**Rating:** 3
**Confidence:** 4

**Summary:**

The paper introduces a novel framework for graph coarsening, Adaptive and Heterogeneous-Universal Graph Coarsening (AH-UGC), that addresses limitations in existing approaches. Traditional graph coarsening techniques often produce a single fixed coarsened graph, are inefficient for multiple target resolutions, and struggle with heterogeneous graphs involving different node and edge types. AH-UGC combines Locality-Sensitive Hashing (LSH) with Consistent Hashing (CH) to efficiently produce multiple coarsened graphs adaptively without redundant recomputation. Furthermore, it incorporates type-isolated coarsening to preserve semantic integrity in heterogeneous graphs by ensuring nodes of different types are not merged.

Contributions:
The authors propose AH-UGC, a unified framework for Adaptive and Heterogeneous Universal Graph Coarsening which integrate locality-sensitive hashing (LSH) with consistent hashing (CH). This framework supports adaptive, streaming, expanding, heterophilic, and heterogeneous graphs.

**Questions:**

1. What is the motivation and contributions of this paper? I hope authors can explain a bit clear since I am a little confused about the background of the problems addressed in the paper. As I mentioned in the weakness, I think it is only a writing problem and I hope it can help me understand the paper correctly.
2. In the problems formulation section, the goal 2 is to learn a coarsening matrix. But the learning details is not clear in section 3.2 (if I understand correctly that section 3.2 is for it).
3. The proofs for the theorems are unclear. In Appendix D, it appears that the authors did not provide a proof for Theorem 3.1 and instead proved a theorem that seems unrelated.
4. What is the significance of Lemma 1? Its purpose and role within the paper are unclear.
5.Which conclusions do the experiments support? This is somewhat unclear and leaves me a bit confused.

**Ethical Concerns:**

["NO or VERY MINOR ethics concerns only"]

**Final Justification:**

After discussion, the authors revised the paper and improve it which make people can understand better the motivavtion and claims. However, I think the novelty is relative weak since it only borrows some basic ideas from gnn domain and lack of insight.

**Limitations:**

Authors don't address the limitations of their work. I suggest authors respond my questions above.

**Quality:**

2

**Strengths And Weaknesses:**

Strengths:
1. The proposed framework, AH-UGC, implements type-isolated coarsening by grouping nodes based on their types and performing coarsening separately within each group. This approach prevents the merging of nodes and edges with incompatible types, thereby preserving the semantic integrity of the original heterogeneous graph.
2. AH-UGC is well-suited for streaming or dynamic graph scenarios, where new nodes and edges are continuously introduced. The proposed method efficiently integrates new nodes into the existing coarsened structure with minimal recomputation.

Weaknesses:
1. The paper is not well-written. It is a little difficult to understand the paper clearly which including motivation and the key contributions of the paper.
2. The overall structure of the paper appears to lack clarity. While the authors present the addressed problems as two distinct goals, only one goal seems to be clearly pursued throughout the paper. Additionally, the meaning and significance of the main theorems, as well as the logical flow of their proofs, are not clearly conveyed. Furthermore, it is unclear which specific conclusions the experimental results are intended to support.

---

> ### Author Rebuttal · Authors · 2025-07-30
>
> We thank the reviewer for their valuable comments and insights and for taking the time to go through our paper.
>
> **Q1)** The paper is not well-written. It is a little difficult to understand the paper clearly which including motivation and the key contributions of the paper.
>
> **Ans)**  We thank the reviewer for the feedback and fully appreciate the importance of clarity in presentation.
>
> To provide context, the flow of the introduction begins by discussing the landscape of homogeneous vs. heterogeneous graphs, and the growing need for scalable solutions due to rapidly increasing graph sizes. We introduce graph coarsening as a practical remedy for scalability, but highlight two key limitations in existing methods:
> (1) lack of adaptive reduction, and
> (2) lack of support for heterogeneous graphs.
>
> These points lead directly to the motivation for our work. From lines 54–67, we clearly enumerate our core contributions, which include:
>
> * A unified, adaptive coarsening framework that uses Locality-Sensitive Hashing (LSH) and Consistent Hashing (CH) to enable multi-resolution, one-pass coarsening — eliminating the need to re-run the algorithm for different compression levels.
> * A heterogeneity-aware extension that introduces type-isolated coarsening to handle multi-typed nodes and edges, enabling compression of heterogeneous graphs without mixing semantics or feature dimensions.
> * In addition, our framework naturally supports streaming settings, heterophilic graphs, large-scale graphs, and dynamically growing structures, making AH-UGC a general-purpose and scalable solution across a wide range of graph modalities.
>
> In the background section, we define key concepts related to graphs and heterogeneous graphs, and formulate two explicit goals tied to the aforementioned challenges. The Related Work section then analyzes the limitations of existing methods, with Figure 2 summarizing the capabilities of prior work and visually situating our contributions.
>
> We hope this clarifies the structural intent and narrative of the paper. That said, we remain open to revising the introduction and background sections further to improve clarity and accessibility in the final version.
>
> **Q2)** The overall structure of the paper appears to lack clarity. ......, it is unclear which specific conclusions the experimental results are intended to support.
>
> **Ans)**
>
> ### **Clarifying Goal Coverage**
> 1) Goal 1 (Adaptivity) is addressed in Section 3.1, with theoretical support from Theorem 3.1 and Lemma 1, and experimental validation in:
>     * Table 1 (coarsening time for multiple ratios)
>     * Table 2 (spectral property preservation)
>     * Table 3 (node classification performance)
>     * Figure 3 (probabilistic guarantees of projection proximity)
> 2) Goal 2 (Heterogeneous Graph Coarsening) is tackled in Section 3.2 via type-isolated coarsening, supported by:
>     * Table 4 (node classification accuracy on heterogeneous datasets)
>     * Figure 4 (supernode purity)
>     * Figure 5 (accuracy vs. coarsening ratio under type-aware settings)
>
> While Goal 2 occupies fewer lines in the paper, it includes a complete algorithm, theory, and empirical evaluation. If the reviewer feels additional elaboration would improve clarity, we are happy to revise and expand those sections accordingly.
>
> ### **Regarding Theorems and Lemma Interpretation**
> In Section 3.1, we use LSH and Consistent Hashing to project high-dimensional augmented features.
> * Theorem 3.1 states that for two nodes x and y that are close in feature space, their hash projections h(x) and h(y) will also be close with high probability:
> $$h(x) = floor((⟨r, x⟩ + b)/w), h(y) = floor((⟨r, y⟩ + b)/w)$$
> * Lemma 1 builds on this by showing that the probability of a distant third node z falling between h(x) and h(y) is very low. This supports our merging strategy by formally bounding the intrusion of dissimilar nodes in merge regions.
>
> These theoretical results draw upon and generalize classical results from LSH theory (e.g., Datar et al., Indyk and Motwani). We’ve cited these at lines 56, 155, and 175 and will further clarify their role in Section 3.
>
>
> **Empirical Validation:**
>
> To support these theoretical claims, we conducted an ablation study where each node is merged with its $k^{th}$ rightmost neighbor (for k=1 to 5). The table below reports node classification accuracy on four benchmark datasets using the coarsened graph:
> | k | Cora   | DBLP | Citeseer | CS|
> |---|--|--|--|--|
> | 1|73.11 |82.33|66.16|87.32|
> | 2| 70.53|82.33|56.30|72.13|
> | 3|69.98 |81.43|52.25|57.70|
> | 4| 63.35|79.45|52.85|58.80|
> | 5|61.51 |75.28|52.40|49.37|
>
> As seen, the performance degrades monotonically with increasing k, validating that:
> * When k=1, we are most likely merging semantically similar nodes—consistent with Theorem 3.1.
> * As k increases, we begin merging less similar nodes, and accuracy declines—supporting Lemma 1, which predicts that further neighbors are less likely to preserve semantic proximity in projection space.
>
> ### **Regarding Experimental Design and Its Purpose**
>
> Each experimental section was designed to directly validate the two goals:
>
> Goal 1: Adaptive Coarsening Support
>
> * Table 1: Validates efficiency and scalability across coarsening ratios.
> * Table 2: Evaluates spectral preservation of coarsened graphs.
> * Figure 3: Empirically validates Theorem 3.1’s claim that projected distances preserve local neighborhoods.
> * Table 3: Shows practical utility by training on coarsened graphs and testing on original ones — a key validation of coarsening quality.
>
> Goal 2: Heterogeneous Graph Coarsening
>
> * Table 4: Demonstrates AH-UGC’s effectiveness on heterogeneous graphs compared to baselines.
> * Figure 4: Shows how AH-UGC avoids impurity in supernodes by respecting node types.
> * Figure 5: Analyzes how downstream accuracy behaves with increasing compression — clearly showing graceful degradation in AH-UGC vs. sudden collapse in others. This also highlights the trade-off between graph size reduction and information retention, which is dataset-dependent.
>
> We hope this clarifies the logical flow from problem motivation to design, theory, and experiments.
>
> **Q3)** What is the motivation and contributions of this paper?.......I hope it can help me understand the paper correctly.
>
> **Ans)** Please refer to Answer 1)
>
> **Q4)** In the problems formulation section, the goal 2 is to learn a coarsening matrix. But the learning details is not clear in section 3.2 (if I understand correctly that section 3.2 is for it).
>
> **Ans)** Indeed, Goal 2 involves learning a coarsening matrix C that satisfies the structural and semantic constraints introduced in lines 111–112.
>
> To handle heterogeneous graphs, we adopt a type-isolated approach (lines 213–218), where the graph is partitioned into subgraphs based on node types. On each type-specific subgraph, we apply the AH-UGC framework described earlier (Lines 152–174) to compute the respective coarsening matrix C, as formalized in Lines 192–200.
>
> This results in a set of coarsening matrices — one per node type — which are then used to reconstruct the heterogeneous coarsened graph. To avoid redundancy, we kept the explanation in Section 3.2 concise, assuming familiarity with the shared logic described in Section 3.1.
>
> We appreciate the reviewer pointing out that this connection could be made more explicit and will revise Section 3.2 to better highlight how coarsening matrices are computed per type and how they contribute to the final heterogeneous graph construction.
>
> **Q5)** The proofs for the theorems are unclear. In Appendix D, it appears that the authors did not provide a proof for Theorem 3.1 and instead proved a theorem that seems unrelated.
>
> Ans)
> ### Theorem 3.1 (Projection Proximity for Similar Points)
>
> Let $x, y \in \mathbb{R}^d$, and define the projection function:
>
> $$
> h(x) = \sum_{j=1}^\ell r_j^\top x, \quad \text{where } r_j \sim \mathcal{N}(0, I_d) \text{ i.i.d.}
> $$
>
> Then the difference $h(x) - h(y)$ follows:
>
> $$
> h(x) - h(y) \sim \mathcal{N}(0, \ell \|x - y\|^2)
> $$
>
> and for any $\varepsilon > 0$:
>
> $$
> \Pr\left[ |h(x) - h(y)| \leq \varepsilon \right] = \operatorname{erf} \left( \frac{\varepsilon}{\sqrt{2\ell} \|x - y\|} \right)
> $$
> ### Proof:
> Let $z = x - y \in \mathbb{R}^d$. Then:
>
> $$
> h(x) - h(y) = \sum_{j=1}^\ell r_j^\top x - \sum_{j=1}^\ell r_j^\top y = \sum_{j=1}^\ell r_j^\top (x - y) = \sum_{j=1}^\ell r_j^\top z
> $$
>
> Each term $r_j^\top z$ is the projection of a standard Gaussian vector onto a fixed vector $z$, so:
>
> $$
> r_j^\top z \sim \mathcal{N}(0, \|z\|^2) = \mathcal{N}(0, \|x - y\|^2)
> $$
>
> Since the $r_j$ are independent, their sum is distributed as:
>
> $$
> h(x) - h(y) \sim \mathcal{N}(0, \ell \|x - y\|^2)
> $$
>
> Now consider the probability:
>
> $$
> \Pr\left[ |h(x) - h(y)| \leq \varepsilon \right]
> $$
>
> This is the cumulative probability within $\varepsilon$ of a zero-mean Gaussian with variance $\ell \|x - y\|^2$. Let $\sigma^2 = \ell \|x - y\|^2$, then:
>
> $$
> \Pr\left[ |Z| \leq \varepsilon \right] = \operatorname{erf} \left( \frac{\varepsilon}{\sqrt{2\sigma^2}} \right) = \operatorname{erf} \left( \frac{\varepsilon}{\sqrt{2\ell} \|x - y\|} \right)
> $$
>
> This completes the proof.
>
> This result gives a formal probabilistic bound showing that closer points in the original feature space are also closer in projected space with high probability — a core property required for our merge strategy to maintain neighborhood consistency.
>
> This theorem is closely aligned with prior LSH theory (e.g., Indyk & Motwani, Datar et al.), but adapted for summed projections, which is specific to our use of consistent hashing over LSH outputs.
>
> **Q6)** What is the significance of Lemma 1? Its purpose and role within the paper are unclear.
>
> **Ans)**  Please see **Regarding Theorems and Lemma Interpretation** in Answer 2.
>
> **Q7)**. Which conclusions do the experiments support? This is somewhat unclear and leaves me a bit confused.
>
> **Ans)** Please see **Regarding Experimental Design and Its Purpose** in Answer 2.

---

> > ### Comment · Reviewer_rETW · 2025-08-03
> >
> > Thanks for authors' effort. Your responds solve parts of my concerns. I will keep my rating.

---

> > > ### Author Response · Authors · 2025-08-03
> > >
> > > Thank you for your comment.
> > >
> > > We have made a *sincere effort to address all the concerns you previously raised*, and we were hopeful that our responses would fully resolve them. In your recent comment, **you indicated that we have only partially resolved your concerns, but you did not specify which aspects are still unclear or unsatisfactory**. We kindly request your help in identifying the remaining issues so we can address them thoroughly while the **discussion phase is still ongoing**. Your detailed feedback would also help us improve the clarity and presentation of the paper more broadly.
> > >
> > > We would also like to respectfully point out that **the other reviewers have found the presentation to be clear** and well-structured.  For instance:
> > >
> > > "Reviewer RpVB: The exposition is quite clear and lucid.
> > >
> > > Reviewer PQfo: The paper is clearly written and easy to follow.
> > >
> > > Reviewer uJX8: The paper is well written and clearly structured."
> > >
> > > This leaves us somewhat uncertain about which specific parts you found confusing. Understanding your perspective in more detail would be extremely helpful for us to identify any overlooked issues and further improve our submission. We truly value your input and remain committed to strengthening the paper based on your guidance.

---

> > > > ### Comment · Reviewer_rETW · 2025-08-06
> > > >
> > > > My concerns are that how those two theorems linked with the clain in your paper? For the proof of the theorems, I think it is standard and nothing novelty. I care about what is the connection between theorecial part and practical part. Maybe show me a concrete example is better.

---

> > > > > ### Author Response · Authors · 2025-08-06
> > > > >
> > > > > We sincerely thank the reviewer for pointing out the need to clarify how **Theorem 3.1** and **Lemma 1** relate to the **practical design and performance** of our coarsening method. Below, we explain this connection and support it with both a concrete example and empirical evidence.
> > > > >
> > > > > ---
> > > > >
> > > > > ### Theoretical Foundation for Our Merge Rule
> > > > >
> > > > > Our coarsening framework uses **consistent hashing** to guide node merging. Specifically, each node is asked to merge with its **nearest right-side neighbor** in hash space. The merging strategy is grounded in the following two results:
> > > > >
> > > > > - **Theorem 3.1**: If two nodes `x` and `y` are **close in feature space**, then with high probability, their hash projections are also close:
> > > > >
> > > > > $$h(x) = floor((⟨W, x⟩ + b)), h(y) = floor((⟨W, y⟩ + b))$$
> > > > >
> > > > >
> > > > > - **Lemma 1**: The **probability that a distant node `z`** intrudes between `h(x)` and `h(y)` is **very low**, assuming `x` and `y` are close and `z` is far from them in the original feature space.
> > > > >
> > > > > ---
> > > > >
> > > > > ### Connection Between Theoretical Results and Coarsening Quality
> > > > >
> > > > > These results directly inform and justify our design choice:
> > > > >
> > > > > - **Theorem 3.1** ensures **locality preservation**: nodes that are semantically similar remain nearby after hashing.
> > > > > - **Lemma 1** guarantees that **dissimilar nodes are unlikely to interfere**, maintaining semantic purity in the merge operation.
> > > > >
> > > > > Thus, when a node is asked to **look to its right and merge**, it is likely to find a similar neighbor and **not a random or noisy one**. This mechanism improves the quality of the coarsened graph, as it ensures that **only similar nodes are merged** into supernodes.
> > > > >
> > > > > ---
> > > > >
> > > > > ### Empirical Validation
> > > > >
> > > > > We validate this connection with an ablation experiment, where each node is forced to merge with its **k-th rightmost neighbor** after projection (`k = 1` to `5`). The table below reports classification accuracy on the coarsened graphs:
> > > > >
> > > > > | Next Neighbor Merge | Cora  | DBLP  | Citeseer | CS    |
> > > > > |---------------------|-------|-------|----------|-------|
> > > > > | 1                   | 73.11 | 82.33 | 66.16    | 87.32 |
> > > > > | 2                   | 70.53 | 82.33 | 56.30    | 72.13 |
> > > > > | 3                   | 69.98 | 81.43 | 52.25    | 57.70 |
> > > > > | 4                   | 63.35 | 79.45 | 52.85    | 58.80 |
> > > > > | 5                   | 61.51 | 75.28 | 52.40    | 49.37 |
> > > > >
> > > > > As seen:
> > > > >
> > > > > - When `k = 1`, nodes merge with their nearest neighbor—achieving highest accuracy.
> > > > > - As `k` increases, nodes merge with farther neighbors—reducing semantic alignment and causing accuracy to decline.
> > > > >
> > > > > These practical results align with Theorem 3.1 and Lemma 1.
> > > > >
> > > > > ---
> > > > >
> > > > > ### Example
> > > > >
> > > > > Suppose three nodes have features:
> > > > >
> > > > > - `x = [0.7, 0.8]`
> > > > > - `y = [0.5, 0.7]` (very close to `x`)
> > > > > - `z = [1.1, 2.2]` (far from both)
> > > > >
> > > > > Let `W = [2.3, 3.1]`, `b = 1`.
> > > > >
> > > > > Then:
> > > > >
> > > > > $$h(x) = floor(((0.7 \times 2.3 + 0.8\times3.1) + 1)), \rightarrow floor(5.09) = 5 $$
> > > > >
> > > > > $$h(y) = floor(((0.5\times2.3 + 0.7\times3.1) + 1)), \rightarrow floor(4.32) = 4 $$
> > > > >
> > > > > $$h(z) = floor(((1.1\times2.3 + 2.2\times3.1) + 1)), \rightarrow floor(10.35) = 10 $$
> > > > >
> > > > > Here, we can see hash values assigned to x, y, z are 5, 4, 10.
> > > > > This guarantees that:
> > > > > * Feature-similar nodes are clustered together.
> > > > > * Dissimilar nodes do not interfere with local merges.
> > > > > * Therefore, merging to the right neighbor (as used in our framework) maintains coarsening quality and local feature consistency.

---

> > > > > > ### Comment · Reviewer_rETW · 2025-08-06
> > > > > >
> > > > > > Thanks for your effort. I will revise my rate.

---

> > > > > > > ### Author Response · Authors · 2025-08-06
> > > > > > >
> > > > > > > Thank you for your **positive support and thoughtful engagement with our work.** As the reviewer–author discussion period nears its conclusion, **we sincerely hope for your support in shaping the final decision on our submission.** We appreciate your time and consideration.
> > > > > > >
> > > > > > > Best,
> > > > > > >
> > > > > > > Authors

---

### Official Review · Reviewer_uJX8 · 2025-06-28

**Clarity:** 4
**Significance:** 2
**Originality:** 2
**Rating:** 3
**Confidence:** 4

**Summary:**

This paper proposes a novel graph coarsening method based on locality-sensitive hashing. The approach applies random projections on the designed node feature vectors, and the coarsening process is then performed based on the aggregated outcomes of these projections. A key advantage of the method is its adaptability. It allows different coarsening ratios without requiring recomputation of the full coarsening pipeline. The authors provide theoretical insights to analyze which node pairs are more likely to be merged. The proposed method is evaluated on several different datasets, demonstrating competitive or superior performance against several baselines.

**Questions:**

**Questions**
- The heterophily factor in Line 147 requires node labels to compute the vectors $F_i$​, but it's unclear how this is handled in scenarios where node labels are unavailable or partially known. If label information is used during coarsening but then withheld during node classification, this may create an unfair advantage in the downstream tasks compared to other methods. Please clarify this point.
- The current node merging strategy relies on sorting nodes by their final hashing scores. However, due to the randomness inherent in LSH, structurally dissimilar nodes can have close similarity scores (node attributes might also be impactful) so it is unclear why this approach should be preferred over more structure-aware alternatives.
- Is the proposed coarsening approach specifically designed for node classification tasks, or is it also applicable to other downstream tasks such as link prediction? If the latter, could the authors clarify how well the method preserves structural properties relevant for link prediction performance? A discussion or empirical evaluation in this context would help demonstrate the generality of the proposed framework.
- Lemma 1 is given without any proof or reference. Please consider including the proof or a reference.
- Theoretical results very similar to Theorem 3.1 have been studied in the literature, so please consider citing relevant works such as [1].
- For the runtime comparisons in Table 1, do the reported times represent averages across all coarsening ratios or cumulative totals?
- In Tables 2 and 3, spectral preservation and classification accuracy are only reported for a 50% coarsening ratio. Why was this specific value chosen, and how does performance vary with different coarsening levels?
- Are the reported results based on a single run or averaged over multiple trials? How does the performance distribution look across different runs? How is the variation for different runs?
- What is the dimensionality of the projection space used in the hashing step? Have you observed any sensitivity to this hyperparameter in terms of performance or stability?
- While the paper claims that the proposed method can be applied to various graph types (e.g., streaming or dynamic graphs), the experiments focus only on node-attributed and knowledge graphs. Authors might provide empirical or theoretical justification to show its applicability to dynamic/streaming scenarios.
- The caption of Table 1 refers to ratios, and according to the definition given in Line 172, they should lie between 0 and 1, but this is not the case. Could you please clarify it?

[1] Matoušek, Jiří. "On variants of the Johnson–Lindenstrauss lemma." Random Structures & Algorithms 33.2 (2008): 142–156.

**Additional Comments**
- Line 38 contains an incomplete sentence: it lacks a verb.
- In Definition 2.2, the meaning of "target labels y" is unclear, as it has not been defined prior to this point.
- Definitions 2.1 and 2.2 use inconsistent notation: in one case, the second entry denotes an adjacency matrix, and in another, it seems to represent the edge set.
- In the appendix, the proof of Theorem 3.2 is given before the proof of Theorem 3.1

**Ethical Concerns:**

["NO or VERY MINOR ethics concerns only"]

**Final Justification:**

After reviewing the authors' clarifications and considering the feedback from other reviewers, I increased my initial score, but I would also be comfortable with a rejection if the AC and the other reviewers find that the paper does not meet the acceptance criteria.

**Limitations:**

The paper does not provide a clear discussion of the limitations of the proposed method. For instance, it is not evident whether the approach can be extended to downstream tasks beyond node classification. Addressing this point, along with the other potential limitations outlined in the Questions section, would strengthen the overall contribution and help clarify the scope of the method's applicability.

**Quality:**

2

**Strengths And Weaknesses:**

**Strengths**
- The paper is well written and clearly structured.
- Empirical results show competitive or superior performance against baselines.

**Weaknesses**
- Several experimental details are either unclear or missing.
- Certain assumptions made in the theoretical results are not clearly stated or justified, and some theoretical claims (e.g., Lemma 1) lack complete proofs.
- The paper proposes a limited novelty.

---

> ### Author Rebuttal · Authors · 2025-07-30
>
> We thank the reviewer for their valuable comments and insights and for taking the time to go through our paper. Due to space limits, we refer to responses given to other reviewers for some similar concerns.
>
> **W1)** Several experimental details are either unclear or missing.
>
> **Ans)** The experimental setup and details are provided in Appendix E. The configurations and implementation details are summarized in Table 7, and when applicable, we cite existing works for architectures adopted directly (see lines 554–558).
>
> That said, we appreciate the suggestion and are happy to add a brief discussion in Appendix E to summarize these details more explicitly, if recommended.
>
> **W2)** Certain assumptions ....(e.g., Lemma 1) lack complete proofs.
>
> **Ans)** Please see Answer 4.
>
> **W3)** The paper proposes a limited novelty.
>
> **Ans)** Please see Reviewer PQfo Weakness 1.
>
> **Q1)** The heterophily factor in Line 147 .....
>
> **Ans)** We clarify that node labels are not used during the coarsening process. The heterophily factor (line 147) is computed once beforehand using only the training nodes whose labels are available.
>
> **Q2)** The current node merging ....
>
> **Ans)** We appreciate this concern and clarify that AH-UGC is inherently structure-aware. The input to LSH is not raw node features but an augmented feature vector that blends node attributes with structural context — ensuring that nodes close in this joint space have similar hash scores.
>
> Further, Theorem 3.1 and Lemma 1 provide formal guarantees: nodes that are close in the augmented space are projected close together with high probability, while distant nodes rarely fall between them, preserving locality during merging.
>
> Our theoretical guarantees are supported by empirical results on spectral preservation (Table 2), node classification (Tables 3–4), and performance under aggressive coarsening (Figure 5). Additionally, AH-UGC is fast, adaptive, and general-purpose, scaling well to homophilic, heterophilic, heterogeneous, and streaming graphs, giving it an edge over other exisitng methods.
>
> To further support Theorem 3.1 and Lemma 1, we conducted an ablation study. We kindly refer the reviewer to Reviewer RpVB – Answer 3.
>
> **Q3)** Is the proposed coarsening approach specifically designed for node classification tasks..
>
> **Ans)** AH-UGC is not limited to a specific downstream task. To assess its generality, we conducted experiments on link prediction
>
> Following are the accuracies for link prediction
> | Dataset  | AH-UGC | UCG   | VAN   | Heavy edge| Kron|
> |----------|--------|-------|-------|-------|-------|
> | DBLP     | 88.63  | 87.48 | 89.14 | 88.36 | 88.12 |
> | Pubmed   | 91.84  | 92.78 | 91.81 | 91.45 | 92.05 |
> | Squirrel | 91.15  | 91.09 | 91.03 | 93.45 | 92.41 |
> | Chameleon| 90.17  | 90.96 | 89.45 | 92.41 | 92.84 |
>
> **Q4)** Lemma 1 is given without any proof ....
>
> **Ans)** Thank you for pointing this out. Lemma 1 is adapted from classical results in randomized projections and order statistics—similar results have been derived in prior works on locality-sensitive hashing (e.g., [Datar et al., 2004], [Indyk & Motwani, 1998]). However, for completeness, we now provide a formal proof below
>
> **Lemma 1 (Separation Probability by Far Point)**
>
> Let $x, y, z \in \mathbb{R}^d$  with  $\|x - y\| \ll \|x - z\|$. Then,
>
> $$
> \Pr[h(x) < h(z) < h(y)] \leq \Phi\left( \frac{\|x - y\|}{\sqrt{\ell} \|x - z\|} \right)
> $$
>
> **Proof:**
> We analyze the chance that a far-away point  z  lies between two close points  x and y in the projected order.
>
> Let:
> $$
> a = h(x) = \sum_j r_j^\top x, \quad b = h(y), \quad c = h(z)$$
>
> Define the difference  $d = h(y) - h(x) \sim \mathcal{N}(0, \ell \|x - y\|^2) \)$.
>
> Assume without loss of generality that  h(x) < h(y). Then:
>
> $$
> \Pr[h(x) < h(z) < h(y)] = \Pr[c - a \in (0, d)]
> $$
>
> Since $h(z) - h(x) \sim \mathcal{N}(0, \ell \|x - z\|^2)$, we compute:
>
> $$
> \Pr[0 < h(z) - h(x) < d] = \int_0^d \frac{1}{\sqrt{2\pi \ell \|x - z\|^2}} \exp\left( -\frac{t^2}{2\ell \|x - z\|^2} \right) dt
> \leq \Phi\left( \frac{d}{\sqrt{\ell} \|x - z\|} \right)
> $$
>
> Taking expectation over \( d \), this gives the desired bound.
>
> **Q5)** Theoretical results very similar to Theorem 3.1 have been studied in the literature...
>
> **Ans)** We appreciate the reviewer’s suggestion and will add the proposed citation to appropriately acknowledge related prior work.
>
> **Q6)** For the runtime comparisons in Table 1, do the reported times represent averages across all coarsening ratios or cumulative totals?
>
> **Ans)** The runtimes reported in Table 1 represent cumulative totals across all coarsening ratios, reflecting the total cost incurred by each method to generate multiple coarsened graphs.
>
> **Q7)** In Tables 2 and 3, spectral preservation and classification .....
>
> **Ans)** The 50% coarsening ratio was chosen arbitrarily as a representative midpoint for comparison across methods. The relation between coarsening ratio and accuracy can be seen from the below Table i.e. if we reduce the graph more and more, we start to see a slight decrease in accuracy values. Hence, there will always be a trade-off when it comes to the coarsening ratio and the quality of the reduced graph.
>
> |Ratio| Cite. | Cora  | CS    | DBLP  | Pub.  | Phy.  |
> |-|-|-|-|-|-|-|
> | 30        | 67.21 | 80.60 | 93.02 | 76.16 | 85.65 | 96.12 |
> | 50        | 65.46 | 77.34 | 92.40 | 75.59 | 80.27 | 94.88 |
> | 70        | 61.91 | 75.16 | 88.29 | 74.83 | 78.15 | 92.43 |
>
> **Q8)** Are the reported results based on a single run ..
>
> **Ans)** Please see Reviewer PQfo Answer 2.
>
> **Q9)** What is the dimensionality of ..
>
> **Ans)** As stated in line 154, the projection matrix has shape (d × l). We set l = 1000 in our experiments. While LSH theory suggests that larger l improves generalization, we observed that performance and stability remain consistent for any l > 1000, making it a reliable default across datasets.
>
> **Q10)** While the paper claims that the proposed method can be applied to various graph types (e.g., streaming ...
>
> **Ans)** Thank you for the suggestion. While our main focus in AH-UGC is on adaptivity and heterogeneous graphs, we agree that supporting streaming graphs is equally important. Since AH-UGC builds upon the principles of UGC (which is designed for streaming), our framework naturally supports incremental coarsening without recomputation.
>
> To further support this claim, we have now conducted additional experiments simulating streaming graph settings. Below are the accuracy and coarsening time results across time intervals for 3 datasets:
>
> **Table: Accuracy and Coarsening Time for Streaming Graphs**
>
> | Interval (% of data) | Cora (Acc — Time) | Pubmed (Acc — Time) | Physics (Acc — Time) |
> |-|-|-|-|
> |0–20|65.01-0.21|81.74-0.35|93.67-4.82|
> |20–30|73.43-0.06|82.91-0.09|94.17-0.99|
> |30–40|74.40-0.05|83.24-0.17|93.73-0.94|
> |40–50|80.01-0.09|83.98-0.21|94.59-1.10|
> |50–60|80.60-0.12|84.31-0.24|94.75-1.33|
> |60–70|82.69-0.13|84.51-0.30|94.90-1.37|
> |70–80|86.37-0.15|85.12-0.34|95.31-1.57|
> |80–90|86.00-0.18|85.32-0.35|96.40-1.39|
> |90–100|86.19-0.10|85.49-0.38|96.12-2.17|
>
> This table presents the **accuracy (Acc)** and **coarsening time (Time)** of AH-UGC on 3 benchmark datasets under a **streaming graph setting**. We assume data arrives in **incremental batches** (10–20% of nodes at a time), and coarsening is applied at each time step without recomputing the entire graph.
>
> **Q11)** The caption of Table 1 refers to ratios, .....
>
> **Ans)** We thank the reviewer for catching this. This is indeed a typographical error — the intended coarsening ratios range from 0.55 to 0.10, not from 55 to 10. We will correct the notation in the caption to clearly indicate that the ratios lie between 0 and 1 (e.g., 0.5 instead of 50%).
>
> ### Additional Comments
>
> **1)** Line 38 contains an incomplete sentence: it lacks a verb.
>
> **Ans)** We have modified the line 38 as:
>
> "Most existing graph reduction techniques, including pooling [16], sampling-based [17], condensation [18], and coarsening-based methods [4, 19, 20], aim to compress graph structures while preserving important properties."
>
> **2)** In Definition 2.2, the meaning of ...
>
> **Ans)** Thank you for pointing this out. In Definition 2.2, “target labels y(target_type)” refers to the labels associated with nodes of the target type, i.e., the specific node type we aim to classify. In heterogeneous graphs with multiple node types, it is common to perform supervised tasks (e.g., classification) only on one type of node — such as papers in DBLP or movies in IMDB. The term target_type reflects this focus.
>
> We will revise the definition to make this explanation explicit in the final version.
>
> **3)** Definitions 2.1 and 2.2 use inconsistent notation: in one case, the second entry denotes an adjacency matrix, and in another, it seems to represent the edge set.
>
> **Ans)** We appreciate the reviewer’s observation. The apparent inconsistency arises because Definition 2.2 explicitly introduces two standard representations for heterogeneous graphs:
>
> The entity-based form: $G(V, E, Φ, Ψ)$, where E is the edge list and $Φ, Ψ$ capture node and edge types.
>
> The type-based form: $G({X(node_{type})}, {A(edge_{type})}, {y(target_{type})})$, where edges are represented using adjacency matrices grouped by edge type.
>
> In contrast, Definition 2.1 (homogeneous graphs) uses the form G(V, A, X), where A is a single adjacency matrix. This is consistent with standard practice and reflects the structural differences between homogeneous and heterogeneous graphs.
>
> We will add a clarifying statement to explain that both notations are intentionally used based on the graph type and level of abstraction required.
>
>
> **4)** In the appendix, the proof ..
>
> **Ans)** Thank you for catching this oversight. We will revise the appendix to present the proof of Theorem 3.1 first, followed by Theorem 3.2.

---

> ### Author Response · Authors · 2025-08-03
> **Rebuttal Follow-up and Request for Further Input**
>
> Dear Reviewer uJX8,
>
> Thank you again for taking the time to review our submission and for sharing your thoughtful feedback; it has been invaluable in helping us refine our work. We’ve done our best to address all the raised concerns carefully in the rebuttal, and **we’re very open to further discussion.**
>
> Since we’re midway through the discussion phase, we just wanted to gently check in to see if there are any remaining questions or points of clarification we could help with while there’s still time. *We’d be more than happy to elaborate on any aspect that might need further explanation.*
>
> And if you feel that your concerns have been adequately addressed, *we would be sincerely grateful if you would consider updating your rating* to reflect your current assessment of the work.
>
> Best,
>
> Authors

---

> > ### Author Response · Authors · 2025-08-06
> > **Appeal to Reviewer**
> >
> > Dear Reviewer,
> >
> > Thank you again for taking the time to review our submission and for sharing your thoughtful feedback. As we approach the end of the discussion phase, we wanted to gently follow up to see if there are any remaining concerns that we can help address. We’re more than happy to elaborate on any aspect that may still be unclear.
> >
> > **Your feedback is indeed important for determining the ultimate fate of our work.** If you feel that your concerns have been sufficiently resolved, **we would be sincerely grateful if you might consider revisiting your score to reflect your updated assessment.**
> >
> > Best,
> >
> > Authors

---

> > > ### Comment · Reviewer_uJX8 · 2025-08-06
> > > **Thank you for your response**
> > >
> > > Thank you for the clarifications provided in your response, including the additional experimental results and answers to the questions I raised. I also appreciate the inclusion of the proof for Lemma 1, but as I mentioned earlier in my review, citing an appropriate reference can be sufficient here, as the lemma does not appear to be a novel contribution of the paper, as you noted.
> > >
> > > As Reviewer rETW also noted, I believe the paper needs to be improved to help clarify several key points. Based on the overall technical novelty and theoretical contribution of the paper, I've revised my score.

---

> > > > ### Author Response · Authors · 2025-08-06
> > > >
> > > > Thank you for your **thoughtful and positive response.** We truly appreciate your time, detailed feedback, and willingness to reconsider your score. **We will ensure that all the clarifications, additions, and improvements discussed during the rebuttal are carefully incorporated into the revised manuscript.**
> > > >
> > > > We sincerely look forward to your support in the final decision process and thank you once again for your valuable efforts and constructive engagement.
> > > >
> > > > Best,
> > > >
> > > > Authors

---

### Official Review · Reviewer_PQfo · 2025-07-04

**Clarity:** 3
**Significance:** 3
**Originality:** 2
**Rating:** 4
**Confidence:** 3

**Summary:**

The paper proposes AH-UGC, a new method for graph coarsening that makes large graphs smaller while preserving their structure and meaning. Unlike previous methods, it can 1) adaptively coarsen a graph to different sizes without starting from scratch, and 2) handle heterogeneous graphs by merging only nodes of the same type. It does this by combining two techniques: LSH to group similar nodes, and Consistent Hashing to efficiently merge them. The method is fast, scalable, and works for different types of graphs, including homophilic, heterophilic, and heterogeneous ones. Extensive experiments show that AH-UGC is more efficient and often more accurate than existing methods.

**Questions:**

1. Is it possible to replace random LSH with a learned hashing function? Would this improve performance?
2. How does graph coarsening impact other downstream tasks, such as graph-level prediction?

**Ethical Concerns:**

["NO or VERY MINOR ethics concerns only"]

**Final Justification:**

I thank the authors for their detailed responses. You have convinced me that graph coarsening has a broader range of applications than I had initially considered. The introduction and experiments could benefit from discussion and empirical evidence showing the effectiveness of your method in these areas. While I remain skeptical about its effectiveness for reducing memory requirements compared to other methods, I recognize that the proposed approach could be of interest to the community. I have raised my score to 4.

**Limitations:**

Yes

**Quality:**

2

**Strengths And Weaknesses:**

### Strengths

1. The paper is clearly written and easy to follow.
2. The proposed method is supported by solid theoretical analysis.

### Weaknesses

1. **Similarity to UGC** The proposed method is highly similar to UGC. Like UGC, AH-UGC uses LSH, runs in linear time, and supports heterophilic graphs. The main differences lie in adaptivity and heterogeneous support, but the core framework remains largely the same.
2. **Reliance on Random LSH** The method depends on LSH, which is inherently randomized. It's unclear how robust the results are across different random seeds. Do the authors average over multiple runs or use a fixed seed? Additionally, is there potential to improve performance by learning the hashing function instead of using random projections?
3. **Questionable Effectiveness of Coarsening** The benefits of graph coarsening are unclear. As shown in Tables 3 and 4, applying a 30%-50% coarsening ratio often leads to a drop of over 10% in accuracy. Figure 5 shows that pushing the coarsening ratio beyond 40% causes a sharp decline in performance. If coarsening brings only modest efficiency gains but significantly harms model quality, it's worth questioning whether the tradeoff is worthwhile.

---

> ### Author Rebuttal · Authors · 2025-07-31
>
> We thank the reviewer for their valuable comments and insights and for taking the time to go through our paper.
>
> **Q1)** Similarity to UGC The proposed method is highly similar to UGC. Like UGC, AH-UGC uses LSH, runs in linear time, and supports heterophilic graphs. The main differences lie in adaptivity and heterogeneous support, but the core framework remains largely the same.
>
> **Ans)** While AH-UGC builds on UGC’s core idea of LSH-based coarsening, it introduces two key innovations that substantially extend its capabilities:
>
> * Unlike UGC, which requires recomputing the entire coarsening process for each ratio (via bin-width tuning), AH-UGC introduces consistent hashing that enables multi-resolution coarsening in a single pass, with theoretical guarantees (Theorem 3.1, Lemma 1). This makes AH-UGC fundamentally adaptive and incrementally refinable — a feature UGC lacks.
>
> * AH-UGC is the first to coarsen heterogeneous graphs while preserving type semantics. We introduce type-isolated coarsening and per-type compression strategies (Section 3.2), allowing us to handle varying feature dimensions and avoid type mixing — a critical limitation in UGC.
>
> We also want to clarify that, empirically, AH-UGC is 4–5× faster than UGC when generating multiple coarsened graphs (Table 1), and achieves +20–30% absolute gain in classification accuracy on heterogeneous datasets (Table 4), demonstrating clear practical advantages.
>
> **Q2)** Reliance on Random LSH The method depends on LSH, which is inherently randomized. It's unclear how robust the results are across different random seeds. Do the authors average over multiple runs or use a fixed seed? Additionally, is there potential to improve performance by learning the hashing function instead of using random projections?
>
> **Ans)**
> We address LSH randomness by using multiple independent projectors, whose aggregation ensures robustness across seeds. The results across 4 different seeds are given below for both runtime and node classification accuracies:
>
> Run time experiments (Across 4 seeds [123, 456, 789, 999])
> | Dataset  | Consistent_hash | UGC            | VAN             | VAE           |
> | -------- | --------------- | -------------- | --------------- | ------------- |
> | Cora     | 7 ± 0.96    | 30 ± 5.20   | 19 ± 0.39    | 13 ± 1.15  |
> | Citeseer | 6 ± 1.25     | 28 ± 13.00  | 28 ± 13.80  | 23 ± 1.36 |
> | DBLP     | 20 ± 3.96    | 131 ± 14.18 | 162 ± 12.31 | 138 ± 17 |
>
> Homophilic and Heterophilic datasets accuracy over multiple runs
> | Dataset   | Model     | Consistent_hash | UGC            | VAN            | VAE            |
> |-----------|-----------|-----------------|----------------|----------------|----------------|
> | Physics   | GCN       | 94.91 ± 0.38  |**95.51 ± 0.22** | 94.91 ± 0.09 | 94.87 ± 0.18 |
> |           | GraphSAGE | 94.51 ± 0.27  | 95.21 ± 0.14 | 96.20 ± 0.13 | **96.24 ± 0.33** |
> |           | APPNP     | 95.36 ± 0.22  | 96.01 ± 1.2  | **96.43 ± 0.14** | 96.24 ± 0.17 |
> | DBLP      | GCN       | 80.2 ± 0.77   | 80.07 ± 0.17 | 79.40 ± 0.63 | **80.40 ± 0.97** |
> |           | GraphSAGE | 69.38 ± 0.39  | 71.44 ± 3.12 | 79.63 ± 0.79 | **80.09 ± 0.61** |
> |           | APPNP     | **85.22 ± 0.18**  | 84.36 ± 0.45 | 83.32 ± 0.51 | 83.74 ± 0.62 |
> | Squirrel  | SGC       |**41.10 ± 1.12** | 40.60 ± 1.18 | 31.59 ± 1.29 | 32.65 ± 1.48 |
> |           | MixHop    | **45.20 ± 1.80**  | 45.00 ± 1.93 | 33.21 ± 0.94 | 31.65 ± 1.76 |
> |           | GPR-GNN   | **45.03 ± 0.97**  | 44.41 ± 1.51 | 29.48 ± 0.69 | 28.40 ± 0.64 |
> | Chameleon | SGC       | 58.95 ± 3.02  | **59.43 ± 2.56** | 38.16 ± 2.80 | 51.32 ± 1.49 |
> |           | MixHop    | 57.85 ± 3.37  | **58.90 ± 3.11** | 37.81 ± 5.27 | 50.48 ± 1.88 |
> |           | GPR-GNN   | 53.99 ± 4.07  |**54.04 ± 2.96** | 38.86 ± 3.26 | 46.67 ± 2.02 |
>
> Heterogenous accuracy experiments (5 trails with different model initialization)
> | Dataset | Model Type | VAN                | VAE            | UGC             | HA-UGC              |
> | ------- | ---------- | ------------------ | -------------- | --------------- | ------------------- |
> | IMDB    | HeteroSGC  | 30.53 ± 3.40     | 27.82 ± 0.12 | 49.61 ± 1.19  | **51.46 ± 1.57**  |
> |         | HeteroGCN  | 35.40 ± 0.09     | 36.36 ± 1.09 | 47.84 ± 1.15  | **52.91 ± 0.33**  |
> |         | HeteroGCN2 | 36.13 ± 0.84     | 36.15 ± 1.31 | 44.13 ± 3.26  | **52.58 ± 0.75**  |
> | DBLP    | HeteroSGC  | 28.33 ± 0.00     | 28.33 ± 0.00 | 53.92 ± 7.58  | **56.60 ± 14.55** |
> |         | HeteroGCN  | 32.07 ± 1.41    | 31.08 ± 0.42 | 58.82 ± 9.23  | **63.13 ± 5.94**  |
> |         | HeteroGCN2 | 31.33 ± 1.19     | 31.35 ± 0.74 | 58.18 ± 9.10  | **62.71 ± 2.56**  |
> | ACM     | HeteroSGC  | 74.25 ± 5.00 | 44.66 ± 1.68 | **60.33 ± 2.69**  | 53.82 ± 6.03   |
> |         | HeteroGCN  | 36.33 ± 0.46     | 37.65 ± 1.70 | 39.27 ± 1.57  | **85.16 ± 2.32**  |
> |         | HeteroGCN2 | 36.76 ± 1.19     | 34.64 ± 1.19 | 49.62 ± 15.91 | **84.36 ± 2.26**  |
>
>
> **Q3)** Questionable Effectiveness of Coarsening The benefits of graph coarsening are unclear. As shown in Tables 3 and 4, applying a 30%-50% coarsening ratio often leads to a drop of over 10% in accuracy. Figure 5 shows that pushing the coarsening ratio beyond 40% causes a sharp decline in performance. If coarsening brings only modest efficiency gains but significantly harms model quality, it's worth questioning whether the tradeoff is worthwhile.
>
> **Ans)** We agree that in a few datasets (1–2 cases), aggressive coarsening (e.g., >70%) results in noticeable accuracy drops. However, for most datasets, AH-UGC maintains high performance even at substantial compression — see Tables 3, 8, and 9.
>
> Importantly, Figure 5 shows that AH-UGC degrades gradually, unlike other methods which suffer sharp drops due to supernode impurity. Our type-aware coarsening ensures semantic preservation even under strong compression (graph size reduced by 60%-70%).
>
> Additionally, coarsening offers significant computational benefits (Table 1). In large-scale settings where training on the full graph is infeasible due to memory or time constraints, graph coarsening may be the only practical option.
>
> **Q4)** Is it possible to replace random LSH with a learned hashing function? Would this improve performance?
>
> **Ans)** Thank you for the question. The core goal of AH-UGC is to enable super-fast, training-free graph coarsening. Introducing a learned hash function would defeat this purpose by requiring training, thereby increasing computational cost and losing the key advantage of scalability.
>
> While it is indeed feasible to learn task-specific hash functions, doing so shifts the method closer to condensation-based approaches like GCond, SFGC, or FGCond (see lines 127–129), which are more expensive and often model-dependent. Our design deliberately avoids this to maintain generality and speed.
>
> **Q5)** How does graph coarsening impact other downstream tasks, such as graph-level prediction?
>
> **Ans)** Graph coarsening is especially beneficial in scenarios involving large graphs, where reducing the graph size leads to substantial computational savings without sacrificing structural integrity. For graph-level prediction tasks such as graph classification, the situation is more nuanced. These datasets often consist of multiple small graphs, where the computational benefits of coarsening may be limited. In such settings, applying graph coarsening could potentially result in information loss, leading to degraded classification performance.
>
> Importantly, we emphasize that coarsening is not restricted to node classification alone. We have also successfully employed AH-UGC for link prediction, demonstrating its versatility. The results for link prediction using coarsened graphs are included in the table below:
>
>
> Accuracies for link prediction
> | Dataset  | AH-UGC | UCG   | VAN   | Heavy edge| Kron|
> |----------|--------|-------|-------|-------|-------|
> | DBLP     | 88.63  | 87.48 | 89.14 | 88.36 | 88.12 |
> | Pubmed   | 91.84  | 92.78 | 91.81 | 91.45 | 92.05 |
> | Squirrel | 91.15  | 91.09 | 91.03 | 93.45 | 92.41 |
> | Chameleon| 90.17  | 90.96 | 89.45 | 92.41 | 92.84 |

---

> ### Author Response · Authors · 2025-08-03
> **Rebuttal Follow-up and Request for Further Input**
>
> Dear Reviewer PQfo,
>
> Thank you again for taking the time to review our submission and for sharing your thoughtful feedback; it has been invaluable in helping us refine our work. We’ve done our best to address all the raised concerns carefully in the rebuttal, and **we’re very open to further discussion.**
>
> Since we’re midway through the discussion phase, we just wanted to gently check in to see if there are any remaining questions or points of clarification we could help with while there’s still time. *We’d be more than happy to elaborate on any aspect that might need further explanation.*
>
> And if you feel that your concerns have been adequately addressed, *we would be sincerely grateful if you would consider updating your rating* to reflect your current assessment of the work.
>
> Best,
>
> Authors

---

> > ### Author Response · Authors · 2025-08-06
> > **Appeal to Reviewer**
> >
> > Dear Reviewer,
> >
> > Thank you again for taking the time to review our submission and for sharing your thoughtful feedback. As we approach the end of the discussion phase, we wanted to gently follow up to see if there are any remaining concerns that we can help address. We’re more than happy to elaborate on any aspect that may still be unclear.
> >
> > **Your feedback is indeed important for determining the ultimate fate of our work.** If you feel that your concerns have been sufficiently resolved, **we would be sincerely grateful if you might consider revisiting your score to reflect your updated assessment.**
> >
> > Best,
> >
> > Authors

---

> > > ### Comment · Reviewer_PQfo · 2025-08-06
> > >
> > > I thank the authors for their detailed responses. My questions regarding novelty, LSH, and graph-level prediction have been addressed. However, I still have concerns about the effectiveness of graph coarsening. The primary motivation for graph coarsening appears to be reducing GPU memory usage when training on large graphs. However, AH-UCG often results in significant accuracy degradation at moderate coarsening levels, with losses of 5% accuracy or more in many cases. Moreover, this degradation is often unpredictable. There are existing methods, such as EXACT [1], that reduce memory overhead without causing substantial drops in accuracy. For these reasons, I will maintain my original score of 3.
> > >
> > > [1] https://openreview.net/forum?id=vkaMaq95_rX

---

> > > > ### Author Response · Authors · 2025-08-07
> > > >
> > > > We thank the reviewer for the constructive feedback and appreciate the opportunity to clarify the motivation and impact of our coarsening framework.
> > > >
> > > > We would like to emphasize that **graph coarsening is not solely motivated by GPU memory reduction** during GNN training. Instead, coarsening serves as a **fundamental preprocessing technique** that enables scalable, interpretable, and efficient graph learning in large-scale and dynamic settings. While memory reduction is one practical application, our use of node classification serves as a **proxy task to evaluate the structural quality** of the coarsened graphs.
> > > >
> > > > Regarding the reviewer’s reference to **EXACT**, we agree that there exist architectures that can reduce memory overhead while preserving accuracy. However, the goals and mechanisms differ. **AH-UGC** provides a **topological compression** of the input graph. While there may be cases where models like EXACT offer slightly better accuracy retention, **graph coarsening offers a broader range of advantages**, as highlighted in the literature:
> > > >
> > > > ---
> > > >
> > > > #### Key Benefits of Graph Coarsening and other graph reduction techniques:
> > > >
> > > > 1. **Neural Architecture Search (NAS)**
> > > >    Graph coarsening/reduction reduces dataset size, enabling faster NAS by minimizing the need to train on full large-scale graphs. This accelerates model selection and lowers compute costs. [1]
> > > >
> > > > 2. **Continual Learning**
> > > >    Informative coarsened graphs act as memory-efficient replay buffers that mitigate catastrophic forgetting in continual learning. For example, CaT [2] uses graph reduction for task updates.
> > > >
> > > > 3. **Visualization and Explanation**
> > > >    Smaller graphs are easier to visualize and interpret. Coarsening enables faster and more human-friendly multilevel graph visualization pipelines. [3]
> > > >
> > > > 4. **Privacy Preservation**
> > > >    Reduced graphs offer inherent privacy benefits by obfuscating fine-grained details. Methods like coarsening/sparsification have been shown to approximate differential privacy while preserving utility. [4]
> > > >
> > > > 5. **Graph Data Augmentation**
> > > >    Coarsening at multiple levels produces diverse graph views, useful for augmentation. For example, HARP generates multi-resolution embeddings via progressive coarsening. [5]
> > > >
> > > > 6. **Low-Memory Deployment**
> > > >    Compact coarsened graphs can be used to train or infer with GNNs on memory-constrained devices, facilitating edge deployment and mobile graph learning.
> > > >
> > > > 7. **Coarsening Applications in different domains**
> > > >    Biology: Coarsening has been effectively used to analyze massive single-cell datasets in genomics and cytometry, where full-resolution graphs are computationally prohibitive[6]. Chemisty: By reducing the size of high-fidelity quantum datasets through locality-sensitive hashing, graph coarsening techniques enable the efficient development of accurate ML potentials for complex chemical systems, significantly lowering the cost of quantum chemical simulations. [7]
> > > >
> > > > ---
> > > >
> > > > Due to these wide-ranging benefits, **graph coarsening and other reduction techniques remain an active and evolving area of research**. We refer the reviewer to the comprehensive survey [8] for further details.
> > > >
> > > > Furthermore, these benefits highlight the broader value of coarsening beyond memory reduction for GNN training, and we will include a clearer motivation in the revised manuscript.
> > > >
> > > > Due to space constraints, we acknowledge that the **motivational aspects of coarsening were under-emphasized** in the manuscript, and we will revise the text to better communicate its broader relevance.
> > > >
> > > > [1] Yang, Beining, et al. "Does graph distillation see like vision dataset counterpart?." Advances in Neural Information Processing Systems 36 (2023): 53201-53226.
> > > >
> > > > [2] Liu, Yilun, Ruihong Qiu, and Zi Huang. "Cat: Balanced continual graph learning with graph condensation." 2023 IEEE International Conference on Data Mining (ICDM). IEEE, 2023.
> > > >
> > > > [3] Zhao, Zhiqiang, Yongyu Wang, and Zhuo Feng. "Nearly-linear time spectral graph reduction for scalable graph partitioning and data visualization." arXiv preprint arXiv:1812.08942 (2018).
> > > >
> > > > [4] Dong, Tian, Bo Zhao, and Lingjuan Lyu. "Privacy for free: How does dataset condensation help privacy?." International Conference on Machine Learning. PMLR, 2022.
> > > >
> > > > [5] Zhao, Tong, et al. "Graph data augmentation for graph machine learning: A survey." arXiv preprint arXiv:2202.08871 (2022).
> > > >
> > > > [6] Kataria, Mohit, et al. "A novel coarsened graph learning method for scalable single-cell data analysis." Computers in Biology and Medicine 188 (2025): 109873.
> > > >
> > > > [7] Anmol, et al. "Locality-Sensitive Hashing-Based Data Set Reduction for Deep Potential Training." Journal of Chemical Theory and Computation (2025).
> > > >
> > > > [8] Hashemi, Mohammad, et al. "A comprehensive survey on graph reduction: Sparsification, coarsening, and condensation." arXiv preprint arXiv:2402.03358 (2024).

---

> > > > > ### Author Response · Authors · 2025-08-07
> > > > >
> > > > > ### AH-UGC Advantages over other coarsening methods with Empirical Validation
> > > > >
> > > > > Compared to prior coarsening methods, **AH-UGC is computationally efficient, adaptive to multiple graph types (including heterogeneous graphs), and scales to large graphs** — as discussed in Section 2. These properties are rarely supported simultaneously in prior work.
> > > > >
> > > > > While we acknowledge that **certain datasets may exhibit higher accuracy drop at extreme coarsening levels**, this is **not representative of the majority of cases**. To clarify this, we conducted additional experiments across a range of coarsening ratios.
> > > > >
> > > > > The table below shows that **AH-UGC maintains competitive accuracy even with 90% reduction** (e.g., column ‘0.1’ corresponds to graphs compressed to 10% of their original size), reinforcing its practical utility.
> > > > > |Dataset|Model|0.9|0.8|0.7|0.6|0.5 |0.45|0.4|0.35|0.3|0.2|0.1|
> > > > > |-|-|-|-|-|-|-|-|-|-|-|-|-|
> > > > > |Physics|GCN|95.10|94.83|94.79|94.52|94.52|94.00| 94.13|93.78|93.96|93.80|93.76|
> > > > > ||APPNP|95.72|95.63|95.71|95.57|95.22|95.07|95.03| 94.71|94.42|94.30|94.01|
> > > > > |DBLP|GCN|83.38|83.47|83.49|83.32|78.84 |79.32|79.06|78.78|78.98|78.62|78.60|
> > > > > || APPNP|85.55|85.38|85.24|84.85|84.82 |84.76|84.57|83.66|83.24|82.11|79.85|
> > > > > |CS|GCN|93.46|93.05|92.91|92.69|92.78| 92.23|90.43|91.49|90.29|90.21|89.19|
> > > > > ||APPNP|94.82|94.87|94.49|94.55|94.14|94.03|93.68|93.32|93.27|93.17|93.11|
> > > > > |Citeseer|GCN|69.67|68.92|67.17|66.17|66.02| 66.10|65.89|65.40|64.80|64.41|64.58|
> > > > > ||APPNP|72.07|71.62|70.87|71.02|69.97|69.57|69.37|68.50|68.47|68.20|68.00|
> > > > > |Pubmed|GCN|87.37|88.11|87.58|87.80|87.53|87.34|86.82|86.76|86.26|85.22|84.51|
> > > > > ||APPNP|87.73|87.70|87.55|87.78|87.47|87.47|87.45|87.42|87.35|87.28|87.11|
> > > > >
> > > > > We emphasize that AH-UGC is the first graph coarsening approach designed for heterogeneous graphs. While Figure 5 shows a drop in accuracy—particularly for the HeteroSGC model around 70% reduction—we believe this highlights the importance of better aligning supernode construction with heterogeneous message-passing mechanisms. As this is a first step in a relatively unexplored direction, we see substantial room for improving the coarsened graph construction pipeline, which we aim to investigate in future work.
> > > > >
> > > > > **Appeal to reviewer:** We hope this addresses your concern. If you find our clarification satisfactory, we would be grateful if you could consider updating your score. Given the encouraging feedback from other reviewers, your revised evaluation could have a meaningful impact on the outcome of our submission. We’re happy to answer any additional questions you may have.
> > > > >
> > > > > Best,
> > > > >
> > > > > Authors

---

> > > > > > ### Comment · Reviewer_PQfo · 2025-08-07
> > > > > >
> > > > > > I thank the authors for their detailed responses. You have convinced me that graph coarsening has a broader range of applications than I had initially considered. The introduction and experiments could benefit from discussion and empirical evidence showing the effectiveness of your method in these areas. While I remain skeptical about its effectiveness for reducing memory requirements compared to other methods, I recognize that the proposed approach could be of interest to the community. I have raised my score to 4.

---

> > > > > > > ### Author Response · Authors · 2025-08-08
> > > > > > >
> > > > > > > Thank you for your **thoughtful and positive response.** We truly appreciate your time, detailed feedback, and willingness to reconsider your score. **We will ensure that all the clarifications, additions, and improvements discussed during the rebuttal are carefully incorporated into the revised manuscript.**
> > > > > > >
> > > > > > > We sincerely look forward to your support in the final decision process and thank you once again for your valuable efforts and constructive engagement.
> > > > > > >
> > > > > > > Best,
> > > > > > >
> > > > > > > Authors

---

> > > > > > > > ### Comment · Reviewer_RpVB · 2025-08-08
> > > > > > > > **Response**
> > > > > > > >
> > > > > > > > Thanks for your feedback and the ablation study. I will revise my score by +1.

---

### Official Review · Reviewer_RpVB · 2025-07-06

**Clarity:** 2
**Significance:** 2
**Originality:** 2
**Rating:** 5
**Confidence:** 2

**Summary:**

The paper seeks to empower current Graph Coarsening methods in two ways:
1) do GC in a "dynamic" fashion, allowing the GC ratio to be dynamically adjusted rather than computing new GC from scratch everytime,
2) handle heterogeneous nodes, i.e. different nodes types in a graph.

The main method to achieve these goals is to integrate LSH and CH. Here CH handles the first problem and LSH handles the second problem.

**Questions:**

-

**Ethical Concerns:**

["NO or VERY MINOR ethics concerns only"]

**Final Justification:**

The authors did a good job of addressing my concerns, especially clarifying the role of this work vis-a-vis GNNs.

**Limitations:**

yes

**Paper Formatting Concerns:**

There is less space after section titles, as compared to other submissions that I have reviewed. Please cross check.

**Quality:**

2

**Strengths And Weaknesses:**

Strengths
1) The exposition is quite clear and lucid.
2) The proposed model attains several attributes simultaneously, as compared to existing GC models.
3) The experimental results are quite extensive.

Weaknesses
1) I am not sure about the degree of technical novelty in the paper, in particular regarding the use of consistent hashing.
2) For GNN node classification, the baselines (such as GCN) are too outdated.
3) An ablation analysis would be required to compare the effect of LSH and CH.

---

> ### Author Rebuttal · Authors · 2025-07-31
>
> We thank the reviewer for their valuable comments and insights and for taking the time to go through our paper.
>
> **Q1)** I am not sure about the degree of technical novelty in the paper, in particular regarding the use of consistent hashing.
>
> **Ans)** While AH-UGC builds on UGC’s core idea of LSH-based coarsening, it introduces two key innovations that substantially extend its capabilities:
>
> * Unlike UGC, which requires recomputing the entire coarsening process for each ratio (via bin-width tuning), AH-UGC introduces consistent hashing that enables multi-resolution coarsening in a single pass, with theoretical guarantees (Theorem 3.1, Lemma 1). This makes AH-UGC fundamentally adaptive and incrementally refinable — a feature UGC lacks.
> * AH-UGC is the first to coarsen heterogeneous graphs while preserving type semantics. We introduce type-isolated coarsening and per-type compression strategies (Section 3.2), allowing us to handle varying feature dimensions and avoid type mixing — a critical limitation in UGC.
>
> We also want to clarify that, empirically, AH-UGC is 4–5× faster than UGC when generating multiple coarsened graphs (Table 1), and achieves +20–30% absolute gain in classification accuracy on heterogeneous datasets (Table 4), demonstrating clear practical advantages.
>
> **Q2)** For GNN node classification, the baselines (such as GCN) are too outdated.
>
> **Ans)**
>
> We would like to clarify that the primary goal of AH-UGC is not to introduce a new GNN model, but rather to introduce a more scalable, adaptive and heterogenous graph coarsening method.
>
> The downstream node classification task is used as a proxy to evaluate the quality of the learned structure. To demonstrate that AH-UGC produces model-agnostic coarsened graph, we deliberately use widely adopted and well-understood GNN backbones for different graph modalities.
>
> We would like to emphasize that we have carefully selected GNN models appropriate to different graph modalities:
> * For homophilic graphs, we evaluate five widely-used GNNs:
> GCN, GIN, GAT, GraphSAGE, and APPNP. Please refer to Table 3 (main paper) and Table 8 in Appendix D for results.
> * For heterophilic graphs, we consider another set of five GNNs designed to handle non-homophilic edge patterns:
> GCN-II, SGC, GPRGNN, MixHop, and GAT-JK. Please refer to Table 3 (main paper) and Table 9 in Appendix D for results.
> * For heterogeneous graphs, we use three specialized models tailored for relational and multi-typed graph settings:
> HeteroSGC, HeteroGCN, and HeteroGCN2. Please refer to Table 4 (main paper).
>
> In total, our **evaluation spans 13 different GNN architectures**, carefully chosen to reflect the diversity of real-world graphs across homophilic, heterophilic, and heterogeneous modalities. As suggested by reviewer, we have also incorporated graph transformer-based GNNs in an extended comparison table,
>
>
> | Dataset  | Model      | Consistent_hash | UGC   | VAN   | VAE   |
> |----------|------------|-----------------|-------|-------|-------|
> | DBLP     | Nodeformer | 76.07          | 71.05 | 73.53 | 71.33 |
> |          | SGFormer   | 72.74           | 68.25 | 79.59 | 74.21 |
> | Physics  | Nodeformer | 79.98           | 90    | 90.89 | 49.95 |
> |          | SGFormer   | 92.18           | 93.65 | 94.97 | 94.00    |
> | Squirrel | Nodeformer | 24.90           | 37.89 | 27.97 | 24.51 |
> |          | SGFormer   | 31.20           | 43.65 | 37.66 | 31.43 |
> | Chameleon| Nodeformer | 36.14           | 46.49 | 35.61 | 42.98 |
> |          | SGFormer   | 47.36            | 49.29 | 47.30 | 50.17 |
> | Cornell  | Nodeformer | 65.95          | 57.44 | 19.14 | 70.21 |
> |          | SGFormer   | 48.93           | 31.91 | 51.06 | 59.57 |
>
>
>
>
> **Q3)** An ablation analysis would be required to compare the effect of LSH and CH.
>
> **Ans)** We appreciate the reviewer’s suggestion. In our CH (Consistent Hashing) strategy, once the nodes are projected using LSH projectors, each node is instructed to merge with its k-th rightmost neighbor in the sorted projected space. We empirically analyze the effect of varying this k in terms of downstream node classification accuracy on coarsened graphs (90% coarsening rate).
>
> The table below shows the classification accuracy when nodes are merged with their 1st to 5th rightmost neighbors. The results clearly indicate a monotonic decline in performance as k increases, validating that the quality of the coarsened graph deteriorates with increasing k:
> | Next Neighbor Merge | Cora   | DBLP | Citeseer | CS|
> |---|--|--|--|--|
> | 1                   |73.11 |82.33|66.16|87.32|
> | 2                   | 70.53|82.33|56.30|72.13|
> | 3                   |69.98 |81.43|52.25|57.70|
> | 4                   | 63.35|79.45|52.85|58.80|
> | 5                   |61.51 |75.28|52.40|49.37|
>
> This trend is consistent with our theoretical results:
> * Theorem 3.1 establishes that two nodes that are close in the original feature space are projected close to each other with high probability.
>
> * Lemma 1 further shows that the probability of a distant node being placed between two similar nodes in the projected space is very low.
>
> Hence, when k=1, the merged node pairs are most likely to be semantically similar, resulting in the highest accuracy and best quality coarsened graph. As $k$ increases, we merge less similar nodes, degrading the representational quality of the graph.

---

> ### Author Response · Authors · 2025-08-03
> **Looking forward to your feedback on rebuttal**
>
> Dear Reviewer,
>
> We thank you for the insightful comments on our work. Your suggestions have now been incorporated in our revision, and we are eagerly waiting for your feedback. As the author-reviewer discussion phase is approaching its conclusion, we are reaching out to inquire if there are any remaining concerns or points that require clarification. Your feedback is crucial to ensure the completeness and quality of our work.
>
> Your support in this final phase would be immensely appreciated.
>
> regards,
>
> Authors

---

> ### Author Response · Authors · 2025-08-06
>
> Dear Reviewer,
>
> We thank you for the insightful comments on our work. We will incorporate your suggestions in our revision, and we are eagerly waiting for your feedback. We are especially grateful for your positive evaluation and for assigning a **Borderline Accept** rating, which reflects your recognition of the contributions of our work.
>
> As the author-reviewer discussion phase is approaching its conclusion, we are reaching out to inquire if there are any remaining concerns or points that require clarification. Your feedback is crucial to ensure the completeness and quality of our work.
>
> If you now feel confident in the completeness and quality of the revised submission, we would be grateful if you might consider revisiting your rating. Even a small adjustment at this stage could make a meaningful difference to the outcome, and your support in the final phase would be immensely appreciated
>
> Warm Regards,
>
> Authors

---

> > ### Author Response · Authors · 2025-08-08
> > **Appeal to Reviewer**
> >
> > Dear Reviewer RpVB,
> >
> > Thank you for your insightful comments.
> >
> > We have incorporated all suggestions, **including additional experiments on newer GNN models and an ablation study.**
> >
> > As the author-reviewer discussion phase is nearing its conclusion in a few hours, we would like to politely inquire if you have any outstanding concerns regarding our work. We’d be more than happy to elaborate on any aspect that might need further explanation.
> >
> > And if you feel that your concerns have been adequately addressed, we would be sincerely grateful if you would consider updating your rating to reflect your current assessment of the work. As we received positive responses from other reviewers, your feedback is indeed important for determining the ultimate fate of our work.
> >
> > Sincerely,
> >
> > Authors

---

### Author Response · Authors · 2025-08-09
**Summary of rebuttal**

We thank the reviewers for their valuable insights and constructive suggestions. Below, we provide a consolidated summary of the key experiments, clarifications, and improvements made in response to their feedback.

### **Summary of Experiments & Clarifications Provided**

- **Novelty & Technical Contributions:**
  1. AH-UGC introduces **Consistent Hashing** for adaptive, multi-resolution coarsening in a single pass, backed by theoretical guarantees.
  2. AH-UGC is the **first coarsening framework for heterogeneous graphs**, ensuring type isolation and preserving semantic integrity during compression.

- **Experimental Breadth:**
  - Evaluated on 23 diverse datasets (homophilic, heterophilic, and heterogeneous).
  - Compared against 13 GNN architectures and Graph Transformer-based models, with additional **newer GNN baselines** included during rebuttal.
  - **Ablation on LSH/CH** demonstrates monotonic accuracy trends with increasing merge distance.
  - **Multi-seed robustness tests** confirm stability across random initializations.
  - **Link prediction experiments** highlight generality beyond node classification.
  - AH-UGC maintains competitive accuracy even with **90% node reduction**.
  - Added results for **streaming graph settings**, showcasing adaptability in dynamic environments.

- **Addressing Reviewer Concerns:**
  - Clarified use of multiple independent LSH projectors to mitigate randomness.
  - Justified the use of random LSH over learned hashing to retain scalability and model-agnostic design.
  - Expanded discussion on broader benefits, including NAS acceleration, continual learning, visualization, privacy preservation, edge deployment, and domain-specific applications.
  - Provided theoretical justification for Theorem 3.1 and Lemma 1, with empirical results validating both.
  - Clarified the overarching motivation and design principles behind AH-UGC.

---

### **Acknowledgement**

We sincerely thank all four reviewers for their active engagement and constructive participation throughout the rebuttal phase. **By the conclusion of discussions, all reviewers agreed that their concerns had been addressed and responded positively to our clarifications, with all raising their evaluations.**

We also extend our gratitude to the Area Chair for their guidance in facilitating a fair, detailed, and productive review process. **We will ensure that all the clarifications, additions, and improvements discussed during the rebuttal are carefully incorporated into the revised manuscript.**

Best Regards,

Authors

---

### Note · Authors · 2025-08-12

We sincerely thank the AC and reviewers for their time and constructive feedback. Your comments have helped us improve both the clarity of our presentation and the depth of our experimental and theoretical validation.

As detailed in our *“Summary of Rebuttal,”* we have addressed all substantive concerns with new experiments, theoretical clarifications, and extended empirical validation.

**By the conclusion of discussions, all reviewers agreed that their concerns had been addressed and responded positively to our clarifications, with all raising their evaluations.** We will ensure that all the clarifications, additions, and improvements discussed during the rebuttal are carefully incorporated into the revised manuscript.

Best Regards,

Authors

---

### Decision · Program_Chairs · 2025-09-17

**Decision:**

Reject

**Comment:**

The paper introduces a fast algorithm for coarsening of heterogeneous heterophilic graphs. The method is simple LSH-based with heuristic width, with a separate handling of heterophilic node features. For the decision, I would side with Reviewers uJX8 and rETW: it seems correct to me to reject the paper for the following reasons: (1) the disconnect of the theory and the actual algorithm: there is almost nothing in the theory that describes the actual algorithm outcome (matrix $C$); (2) experiments in unimportant regimes: there is practically very limited need to reduce the size of a graph by 2x when applying GNNs: training can be done on subgraphs; as can inference.